# Deep learning suggests that gene expression is encoded in all parts of a co-evolving interacting gene regulatory structure

Jan Zrimec [1], Christoph S. Börlin[1,2], Filip Buric [1], Azam Sheikh Muhammad [3], Rhongzen Chen[3], Verena Siewers [1,2], Vilhelm Verendel[3], Jens Nielsen [1,2], Mats Töpel [4,5] & Aleksej Zelezniak [1,6 ✉]

Understanding the genetic regulatory code governing gene expression is an important challenge in molecular biology. However, how individual coding and non-coding regions of the gene regulatory structure interact and contribute to mRNA expression levels remains unclear. Here we apply deep learning on over 20,000 mRNA datasets to examine the genetic regulatory code controlling mRNA abundance in 7 model organisms ranging from bacteria to Human. In all organisms, we can predict mRNA abundance directly from DNA sequence, with up to 82% of the variation of transcript levels encoded in the gene regulatory structure. By searching for DNA regulatory motifs across the gene regulatory structure, we discover that motif interactions could explain the whole dynamic range of mRNA levels. Co-evolution across coding and non-coding regions suggests that it is not single motifs or regions, but the entire gene regulatory structure and specific combination of regulatory elements that define gene expression levels.

[1] Department of Biology and Biological Engineering, Chalmers University of Technology, Kemivägen 10, SE-412 96 Gothenburg, Sweden. [2] Novo Nordisk Foundation Center for Biosustainability, Chalmers University of Technology, Kemivägen 10, SE-412 96 Gothenburg, Sweden. [3] Computer Science and Engineering, Chalmers University of Technology, Kemivägen 10, SE-412 96 Gothenburg, Sweden. [4] Department of Marine Sciences, University of Gothenburg, Box 461, SE-405 30 Gothenburg, Sweden. [5] Gothenburg Global Biodiversity Center (GGBC), Box 461, 40530 Gothenburg, Sweden. [6] Science for Life Laboratory, Tomtebodavägen 23a, SE-171 65 Stockholm, Sweden. ✉email: aleksej.zelezniak@chalmers.se

Gene expression governs the development, adaptation, growth, and reproduction of all living matter. Understanding its regulatory code would provide us with the means to cure diseases, including heterogeneous tumours[1], and to control protein production for biotechnology purposes[2,3]. Although transcriptional regulation has been a central area of research in the past decades, with advances that enable accurate measurement of mRNA levels ranging from just a few copies to several thousand per cell[4–7], we still cannot quantify to what extent the DNA code determines mRNA abundance, nor understand how this information is encoded in the DNA. Lack of such quantitative understanding hinders the potential of accurately controlling mRNA and protein levels by simply manipulating the sequence of the four DNA nucleotides.

A strong agreement between protein and mRNA levels in multiple organisms suggests that transcription is a major determinant of protein abundance[4,5,7,8]. mRNA transcription is controlled via the gene regulatory structure, comprised of coding and cis-regulatory regions that include promoters, untranslated regions (UTRs) and terminators, each encoding a significant amount of information related to mRNA levels[9]. For instance, in Saccharomyces cerevisiae, sequence properties of individual cis-regulatory regions can explain up to half of the variation in mRNA levels[10–15]. Considering that each part of the gene regulatory structure is involved in specific processes related to mRNA synthesis, decay[9,16,17], and the overall transcription efficiency[18], all gene parts must cooperate in perfectly timed concerted action in order to regulate expression. However, despite the apparent importance of the non-coding regions in mRNA control, it remains unclear how they cooperatively regulate gene expression levels.

Much of the current knowledge on quantitative regulation of gene expression is based on high-throughput screens of thousands of synthetic sequences studied in isolation from their native gene regulatory structures[19–22]. Although a de facto standard for expression tuning in synthetic biology, these techniques are (i) laborious and require expensive and highly-sophisticated equipment[23], (ii) are biased towards specificities of mutagenesis[24], and (iii) are generally restricted to particular experimental conditions. The major problem, however, is that the biological sequence space is so large that it cannot be explored experimentally or computationally[25]. For instance, to analyze all the possible combinations of the four nucleotides in a 20 bp promoter would require iterating over a trillion ($4^{20}$) synthetic sequences. This limits the experimental studies to individual regulatory gene parts in the context of single reporter genes. Similarly, with natural systems, the majority of studies on mRNA transcription in the context of transcription factor (TF) binding[26], chromatin accessibility[27,28] and ChIP-seq or DNase-Seq data[29,30], focus solely on promoter regions[22,31]. Therefore, both the current natural and synthetic approaches are fundamentally limited in their ability to study the holistic relationship between the different parts of the gene regulatory structure and their joint regulation of expression.

Here, we consider that DNA sequences of all living systems, through evolution, have been fine-tuned to control gene expression levels. To learn from the natural systems, we analyse over 100,000 native gene sequences in over 20,000 RNA-Seq experiments from seven model organisms, including Homo sapiens and Saccharomyces cerevisiae. In the yeast S. cerevisiae, the variation of gene expression per gene across the entire repertoire of different experimental conditions is 340 times lower than the variation of expression levels across all genes. Consequently, the deep neural networks learn to predict gene expression levels directly from the native DNA sequences, without the need for screening experiments using synthetic DNA. Prediction of gene expression

levels is highly accurate in all model organisms, and in S. cerevisiae ($R^2_{test} = 0.82$) shows strong agreement with fluorescence measurements from independent published experiments. This demonstrates that, in both eukaryotes and prokaryotes, mRNA levels are determined not by separate individual coding and cis-regulatory regions but rather collectively by the entire gene regulatory structure. In S. cerevisiae, as the coding and cis-regulatory regions contain both orthogonal as well as overlapping information on expression levels, the entire gene codon distribution can be predicted merely from the adjacent cis-regulatory regions ($R^2_{test} = 0.58$). Indeed, mutational analysis of orthologous genes in 14 yeast species provides evidence that each gene is a co-evolving unit. Next, we reconstruct the regulatory grammar of S. cerevisiae by measuring the co-occurrence of sequence motifs extracted from the deep models and present across the cis-regulatory regions. These motif co-occurrences are found to be highly predictive of expression levels and can differentiate the expression levels of single motifs in a range of over 3 orders of magnitude. Finally, by quantifying the variation present in all 36 million promoter-terminator combinations we observe that gene expression levels change, on average, over 10-fold in either direction of the native levels. Thus with each gene, merely exchanging one side of its regulatory regions with other natural variants unlocks enormous potential for future gene expression engineering. The potential of our models to guide gene expression engineering with any desired gene in S. cerevisiae is demonstrated experimentally using GFP fluorescence measurements.

## Results

**The dynamic range of gene expression levels is encoded in the DNA sequence.** To explore the relationship between DNA sequence and gene expression levels, we compiled a dataset of 3025 high-quality Saccharomyces cerevisiae RNA-Seq experiments[32] covering the majority of available experimental conditions from 2365 unique studies. By sorting 4975 protein-coding genes ("Methods", Supplementary Table 1) according to their median expression levels across the RNA-Seq experiments (expressed as TPM values, i.e. Transcripts Per Million), we observed a striking trend: expression levels varied within one-fold of values for 79% of the yeast protein-coding genes (Fig. 1a) and within 1 relative standard deviation ($RSD = \sigma/\mu$) for 85% of the genes (Fig. 1a, b and Supplementary Table 2). Conversely, the dynamic range of average TPM values across all the genes spanned over 4 orders of magnitude and the variance of expression levels within the whole genome was on average 340 times higher than the variance per gene across the experiments (Fig. 1c). The most variable genes across the entire range of biological conditions (Fig. 1b: RSD > 1) were significantly (Hypergeometric test BH adj. p-value < 0.05) enriched in metabolic processes, transport and stress response (Supplementary Fig. 2, "Methods"), while the most stable genes (RSD < 1) were significantly (Fisher's exact test p-value < 1e-16) enriched in TFIID-type constitutive promoters[27].

To test if the observed dynamic range of gene expression levels is encoded in the DNA, we extracted the DNA sequences of the regulatory and coding regions of all the genes with a relative standard deviation of expression variation less than 1 (Fig. 1a, b: 4238 genes with RSD < 1). A total of 2150 bp of regulatory sequences[9,11,14,15,33–37], 64 codon frequencies from coding regions[38] and additional 8 mRNA stability variables[13], all known to be important for expression regulation, were used for prediction of mRNA levels (Fig. 1d and Supplementary Fig. 1, "Methods"). Using the DNA sequence information as input, we built a regression model based on deep convolutional neural networks (CNNs), capable of identifying functional DNA motifs

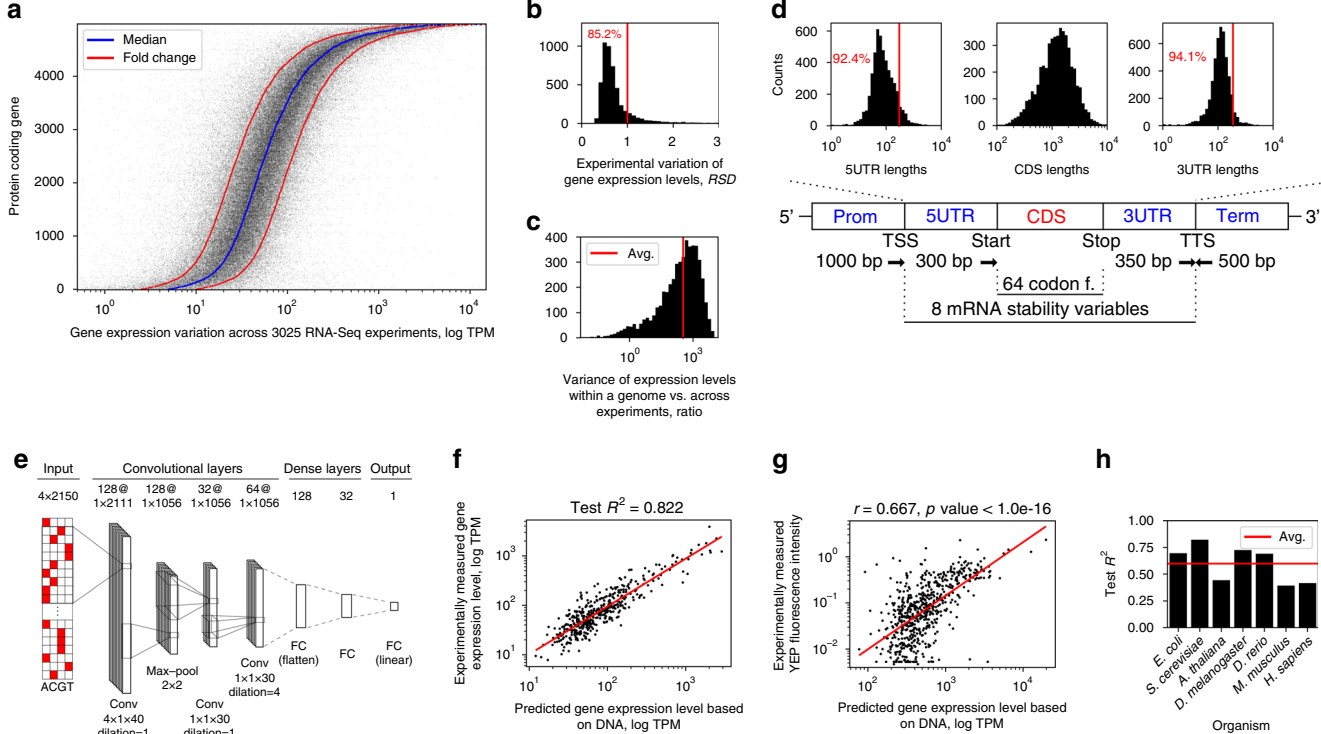

**Fig. 1 The dynamic range of gene expression levels is encoded in the DNA sequence.** All results shown are for *S. cerevisiae*, except in (h). **a** Expression levels (transcripts per million, TPM) of protein-coding genes across 3025 RNA-seq experiments. Inset: distribution of genes with a relative standard deviation ($RSD = \sigma/\mu$) below 1. **b** Experimental variation of the gene expression levels expressed as *RSD*. **c** Distribution of ratios between the variance of expression levels within a genome and the variance of expression levels per gene across the experiments. **d** Schematic diagram of the explanatory variables used for modeling, with distributions of the sequence lengths in the regions where the lengths varied. "TSS" denotes the transcription start site and "TTS" the transcription termination site. **e** The optimized deep neural network (NN) architecture, where "conv" denotes convolutional NNs, "FC" fully connected layers, and "max-pool" max-pooling layers. The values denoting sizes of parameters *layers*, *stride*, *kernels*, *filters*, *max-pooling*, and *dilation* are specified. **f** Experimentally determined (true) versus predicted expression levels with the *S. cerevisiae* model on the held out test dataset (*n* = 425). Red line denotes least squares fit. **g** Comparison of published experimental fluorescence measurements[44] with the predicted expression levels (*n* = 625). Red line denotes least squares fit. **h** $R^2$ on held out test datasets across 7 model organisms. Red line denotes mean value. Source data are provided as a Source data file.

across the sequence[39–42], and trained the model to predict the median gene expression levels (Fig. 1e, "Methods"). The median value was a good estimator of the expression differences between the genes, as it was strongly correlated with the first principal component of the entire expression matrix (Pearson's $r = 0.990$, *p*-value < 2e-16, Supplementary Fig. 3). To avoid potential technical biases related to read-based sequencing[43], mRNA levels were corrected for gene length bias (Supplementary Fig. 4, "Methods"). Overall, a total of 3433 gene sequences were used for training the model, 381 for tuning the model hyperparameters and 424 for testing. After optimizing the model ("Methods"), its predictive performance on the held out test set ($R^2_{\text{test}} = 0.822$, *p*-value < 1e-16, Fig. 1f and Supplementary Table 3) demonstrated that the DNA encodes the majority of the information about mRNA expression levels.

To validate the trained model on independent data, we used two published experimental datasets, where the effects of either promoter[44] or terminator[45] sequences were measured in yeast synthetic constructs combined with fluorescence reporters. For both studies, we only inferred expression levels based on the deep neural network trained on natural genomic sequences (Fig. 1e, f, "Methods"), meaning that the model was not exposed to the data from these studies in its training phase. The first dataset comprised measured activities of ~900 native yeast promoters recorded in synthetic constructs with a single strong terminator (*ADH1*) and a YFP fluorescence reporter in 10 different conditions[44]. In all 10 conditions, the predictions of mRNA

levels inferred based on the DNA sequences of the synthetic constructs and YFP codon frequencies were in strong agreement (Pearson's *r* from 0.570 up to 0.718, *p*-value < 1e-16) with the experimental YFP readouts (Fig. 1g: median YFP readout shown, Pearson's $r = 0.667$, *p*-value < 1e-16, Supplementary Fig. 5a). Similarly, the second experimental dataset[45] contained expression measurements of over 5,000 terminators with a fixed strong promoter (*TDH3*) and a GFP fluorescence reporter. Despite the fact that predictions based on these synthetic constructs were only moderately correlated (Pearson's $r = 0.310$, *p* < 1e-16) with the measured protein fluorescence intensities (Supplementary Fig. 5b), this result was in fact stronger than the correlation between reporter fluorescence and measured mRNA abundances reported in the original study (Pearson's $r = 0.241$)[45].

To investigate if mRNA abundance can be predicted from the DNA sequence also in other model organisms, we processed an additional 18,098 RNA-Seq experiments from 1 prokaryotic and 5 eukaryotic model organisms equally as with yeast (Supplementary Tables 1 and 2, "Methods"). The organisms were selected to cover the whole known range of genome regulatory complexity, from 892 genes/Mbp (*Escherichia coli*) to 6 genes/Mbp (*Homo sapiens*), which is known to affect the gene structure and regulation[46]. After training the models for each organism, the predictive power on test data ($R^2_{\text{test}}$) varied from 0.394 for *Mus musculus* to 0.725 for *Drosophila melanogaster* (*p*-value < 1e-16, Fig. 1h, Supplementary Table 3). Overall, the predictions were less accurate for higher eukaryotes, which could be attributed to the

increase in transcriptional complexity, e.g. due to alternative splicing[47], expression differences across tissues[48], and distant enhancer interactions[49–52], which were not accounted for in the present models. The prediction performance was thus correlated (Pearson's $r = 0.616$, $p$-value < 3e-3) with the genomic complexity of the model organisms (Supplementary Fig. 6). Nevertheless, the average performance across model organisms ($R^2_{test}$) of 0.6 (Fig. 1h) corroborated that the majority of mRNA expression differences in all organisms can be predicted directly from the DNA.

**Coding and cis-regulatory regions jointly contribute to gene expression prediction.** To evaluate the importance of each part of the gene regulatory structure (Fig. 1d) for the prediction of expression levels, we measured the amount of relevant information in each regulatory region. Similarly to the complete model (Fig. 1e, f), we trained multiple CNN models independently on promoter, 5′-UTR, 3′-UTR, and terminator regions as well as their combinations. To justify the use of deep convolutional networks, as a baseline we performed shallow modeling using a variety of regression algorithms, including multiple linear regression, elastic net, random forest, and support vector machines with nested cross-validation (Fig. 2a, "Methods"). We observed that, although a single regulatory region alone could explain less than 28% of the variation in mRNA abundance levels, when using combinations of regulatory regions, each region contributed to the prediction of mRNA levels and increased model performance (Fig. 2b, Supplementary Fig. 11). The model trained on all four regulatory regions thus accounted for approximately 50% of the mRNA abundance variation ($R^2_{test}$ = 0.492, $p$-value < 1e-16, Fig. 2a, b and Supplementary Table 8), suggesting that the entire gene regulatory structure is important for controlling gene expression levels. In contrast, none of the shallow models could predict gene expression levels from the regulatory sequences ($R^2_{test}$ < 0.031, Fig. 2a and Supplementary Table 9), likely since they cannot decode the information in a DNA sequence directly[53,54] and thus rely on human-engineered features, such as k-mer frequencies, as a representation of the sequence properties[55–58]. Despite the observation that the sole mRNA stability variables were also somewhat informative about gene expression levels ($R^2_{test}$ = 0.378, $p$-value < 1e-16, Fig. 2a), they did not improve the overall model performance next to the codon frequencies and regulatory sequences, likely due to their information redundancy with these variables (Supplementary Fig. 12: $R^2_{test}$ was 0.779 when predicting mRNA stability variables using regulatory sequences).

On the other hand, the codon frequencies alone explained well over 50% of mRNA level variation both with the shallow and deep models ($R^2_{test}$ was 0.681 and 0.690, respectively, $p$-value < 1e-16, Fig. 2a, Supplementary Table 9). Considering that the amount of information was comparable to, or greater than that of the regulatory sequences, we attempted to measure the amount of information overlap in these regions. We thus trained a CNN to predict the entire codon frequency vector for each gene using only its regulatory regions (Fig. 2c, Methods), which showed that over 58% of the gene's codon frequency variation was determined by its adjacent regulatory regions (Fig. 2d: $R^2_{test}$ = 0.582, $p$-value < 1e-16). This result suggested that the coding and noncoding regulatory regions might have co-evolved under common evolutionary pressure[59,60]. To test this hypothesis, we composed a dataset of cis-regulatory regions of orthologous genes from a diverse set of 14 yeast species[61] (Supplementary Table 10) and compared the mutation rates between the different regions of the yeast gene structure (Methods). The mutation rates of the promoter and terminator regions displayed a moderate positive correlation

(Pearson's $r = 0.423$ and 0.471, $p$-value < 1e-16, respectively) with the mutation rates of the yeast coding regions (Fig. 2e, f) as well as among themselves (Supplementary Fig. 13), supporting the hypothesis that elements of the gene regulatory structure co-evolve[59,60,62,63].

**Deep learning identifies specific DNA positions controlling gene expression levels.** To explore the information learned by the deep neural network (Fig. 1e, f) and identify the specific parts of the DNA sequences that were most predictive of gene expression levels, we developed a pipeline for evaluating the *relevance* of each specific nucleotide position in relation to the predicted gene expression levels (Supplementary Fig. 14, "Methods"). Briefly, for each gene we removed sliding windows of 10 base pairs along its regulatory DNA sequence (Supplementary Fig. 15) and compared the predictions of the occluded sequences with those of the original unoccluded sequences[39,64]. The occluded parts of the input DNA sequences that significantly deviated (exceeding ±2 standard deviations) from the original data were regarded as the most relevant for gene expression changes (Fig. 3a). The largest density of relevant regions was obtained in the direct vicinity of the boundary sites defining regulatory and coding sequences (see Fig. 1d). On average, 214 base pairs (bp) in promoters, 74 bp in 5′-UTRs, 94 bp in 3′-UTRs, and 127 bp in terminator regions of each gene significantly affected the prediction of its expression levels (Fig. 3a). Relevance profiles of promoter regions were also strongly correlated (Pearson's $r = -0.7$, $p$-value < 1e-16) with experimentally measured nucleosome occupancy scores[28] (Supplementary Fig. 16), which suggested that the deep learning algorithms uncovered the intrinsic molecular information encoded in the nucleotide composition.

Clustering the gene regulatory regions according to their expression *relevance* profiles (Methods) identified 4 stable clusters that significantly (Wilcoxon rank-sum test $p$-value < 1e-4) differed in the positional information (Fig. 3b) and were significantly (Wilcoxon rank-sum test $p$-value < 1e-16) informative about gene expression levels (Fig. 3d). Cluster 4, which contained highly expressed genes, was significantly (Hypergeometric test $p$-value < 1e-10) enriched in the occupied proximal-nucleosome (OPN) regulation strategy as opposed to the depleted proximal-nucleosome (DPN) strategy[65], which was likely related to the concurrent enrichment (Fisher's exact test $p$-value < 1e-12) of inducible SAGA promoters in the cluster. Cluster 4 also comprised genes with higher transcriptional plasticity spanning an over 4-fold higher variability of expression levels compared to the other clusters (Levene's test $p$-value < 1e-16). As is typical for highly abundant proteins, such as metabolic enzymes, that are linked to less defined nucleosome positions and higher dependence on nucleosome remodelling[65,66], the cluster was significantly enriched (Hypergeometric test BH adj. $p$-value < 0.01) mostly in metabolic processes (Supplementary Fig. 17). In contrast, Cluster 1, with lowly expressed genes, was related (Hypergeometric test BH adj. $p$-value < 0.01) to cell cycle regulation and DNA repair (Supplementary Fig. 17). The largest differences in positional expression *relevance* were identified in promoter and terminator sequences (Fig. 3b). For instance, in promoters of lowly and highly expressed genes (Clusters 1 and 4, respectively), occluding the original sequences yielded opposite effects. These positional differences were independent of the overall nucleotide composition (Supplementary Fig. 18), which indicated that specific regulatory DNA motifs were likely responsible for defining the expression levels.

We next identified the specific regulatory DNA motifs important for predicting expression levels from the set of all significantly relevant DNA sequences (Supplementary Fig. 19)

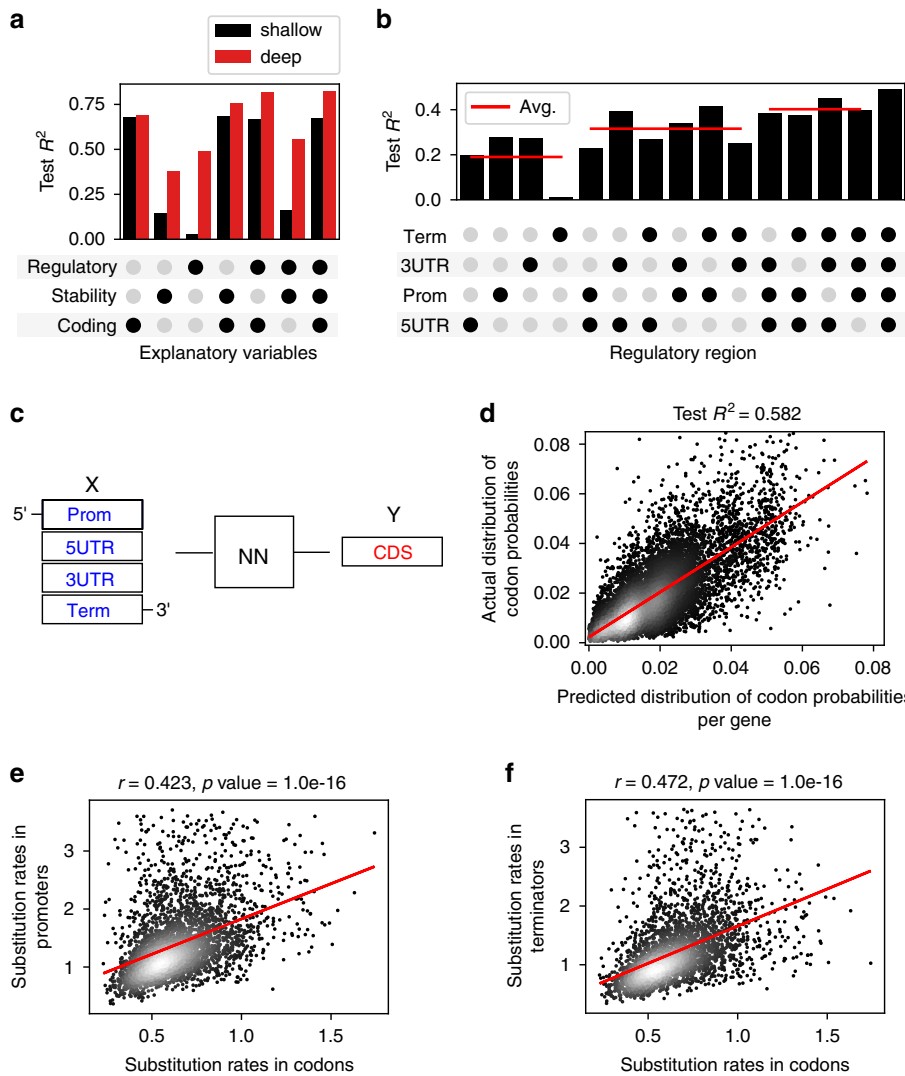

**Fig. 2 Coding and cis-regulatory regions jointly contribute to gene expression prediction. a** Test $R^2$ with different combinations of coding (codon prob.), mRNA stability (see Supplementary Fig. 1c) and cis-regulatory regions, using shallow (black) and deep modeling (red). **b** Test $R^2$ with deep models trained on different combinations of cis-regulatory regions. Red lines denote mean values. **c** Schematic depiction of deep neural network (NN) for prediction of codon probabilities (Y) based on regulatory DNA as input (X). **d** Actual versus predicted distribution of codon probabilities as predicted from non-coding DNA regions based on the model for prediction of codon probabilities based on regulatory DNA as input ($n = 11,947$). Red line denotes least squares fit. **e** Evolutionary substitution rates in promoter vs. coding regions in orthologous genes of 14 yeast species ($n = 3248$). Red line denotes least squares fit. **f** Evolutionary substitution rates in terminator vs. coding regions in orthologous genes of 14 yeast species ($n = 3248$). Red line denotes least squares fit. Source data are provided as a Source data file.

using clustering and alignment ("Methods"). The highest quality motifs were obtained with the 80% sequence identity cutoff, according to the following criteria: (i) genome coverage, (ii) the amount of retained relevant sequences in motifs (seq. coverage) and (iii) % overlap with known motifs in databases (Supplementary Table 11). Over 2200 expression related regulatory DNA motifs were uncovered across all 4 regulatory regions (Fig. 3e). The majority of motifs were unique to each specific region, as analysis of motif similarity across the adjacent regulatory regions (Methods) showed that, on average, <16% of the motifs significantly (Tomtom[67] BH adj. p-value < 0.05) overlapped between the regions (Fig. 3e and Supplementary Table 11). This further supported our observation that every regulatory region contains unique information related to the gene expression levels (see Fig. 2a). Further comparison to JASPAR[68] and Yeastract[69] databases ("Methods") showed that on average, 13% of the

identified motifs (Fig. 3e) were significantly (Tomtom[67] BH adj. p-value < 0.05) similar to the known transcription factor binding sites (TFBS) recorded in these databases (Supplementary Fig. 20), recovering 38% and 63%, respectively. The majority of these motifs were identified not only in the promoter, but also in the terminator region (Supplementary Fig. 20), due to the overlap between neighbouring genes[70]. A significant (Wilcoxon rank-sum test p-value < 1e-8) decrease in GC content in the UTR regions (Fig. 3c) indicated that the identified motifs were likely to contain the regulatory DNA signals from UTR regions and terminators (Fig. 3f and Supplementary Fig. 1). This included the 5′-UTR Kozak sequences and 3′-UTR processing DNA elements that are enriched in the thermodynamically less stable A/T nucleotides[11,14,71,72]. The deep models therefore successfully identified known regulatory signals (Fig. 3f), as well as uncovered new regulatory signals across all the gene regulatory regions[73,74].

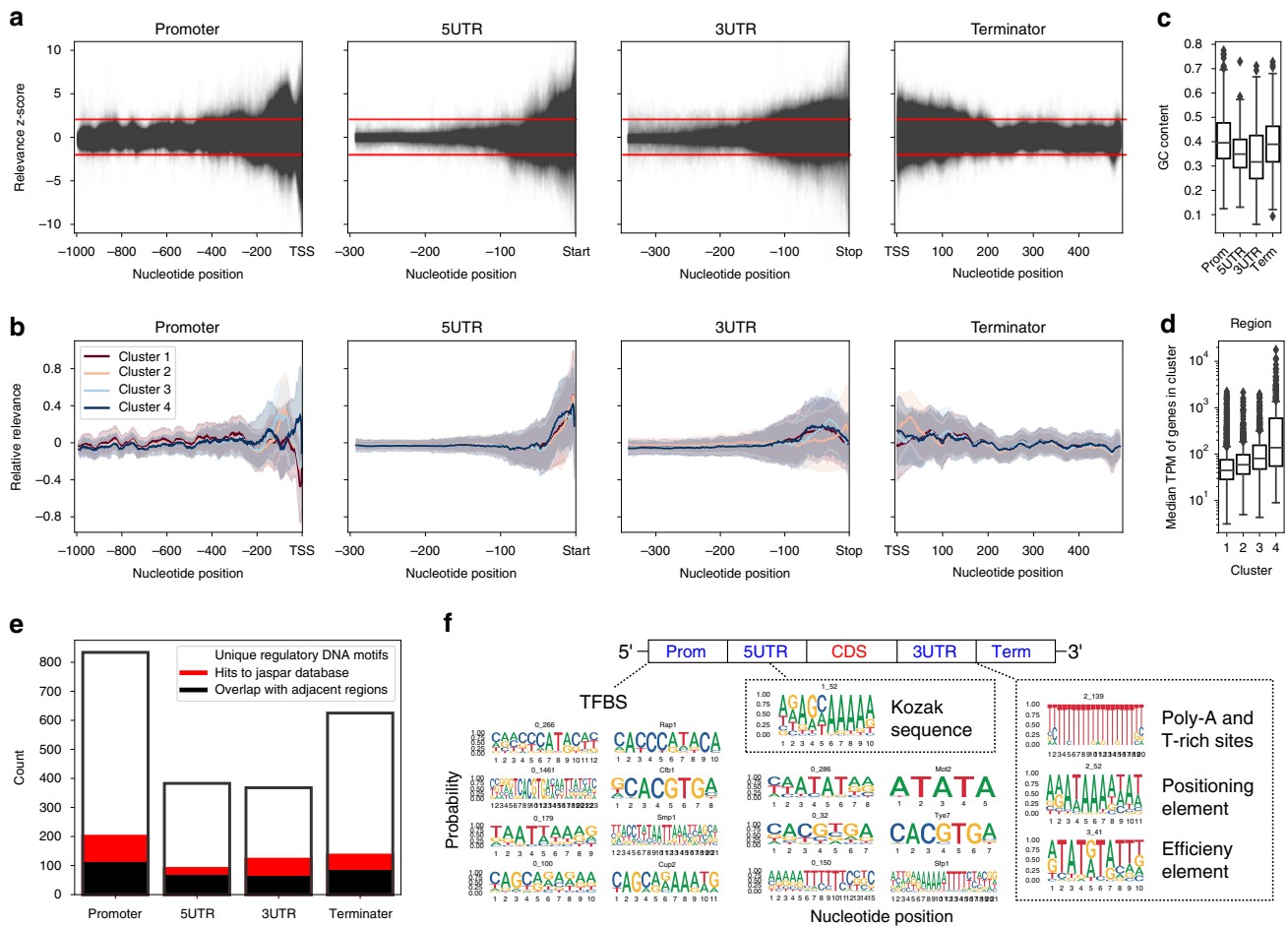

**Fig. 3 Deep learning identifies the DNA positions predictive of expression levels. a** Relevance profiles across *cis*-regulatory region sequences obtained by querying the deep models (Fig. 1f). Red lines denote cutoff of 2 standard deviations. The decrease in relevance of UTR sequences at left edges was due to the considerable number of sequences that were shorter than the analysed regions (see Fig. 1d). **b** Clustered relevance profiles across the *cis*-regulatory region sequences. The clusters 1 through 4 are colored dark red, light red, light blue and dark blue, respectively. Lines and shaded regions represent means and standard deviations, respectively. **c** GC content in the *cis*-regulatory regions (n = 368, 383, 834 and 625, respectively). **d** Median expression levels of genes in the clustered relevance profiles (n = 1385, 1187, 1206 and 460 with increasing cluster size, respectively). **e** Total amount of regulatory DNA motifs uncovered in the *cis*-regulatory regions, with the proportional amounts of motif overlap between adjacent regions (black) as well as with the Jaspar database[68] (red) highlighted. **f** Examples of regulatory DNA motifs uncovered across all the *cis*-regulatory regions that correspond to published motifs and sequence elements (see Supplementary Fig. 1c). "TFBS" denotes transcription factor binding sites. For box plots in **c**, **d**, boxes denote interquartile (IQR) ranges, centres mark medians and whiskers extend to 1.5 IQR from the quartiles. Source data are provided as a Source data file.

**Motif co-occurrence uncovers the regulatory rules of gene expression**. To determine the functional meaning of the DNA motifs reconstructed from the deep learning *relevance* profiles (Fig. 3f), we analysed the informative power of the DNA motifs for predicting the specific gene expression levels. For this, we calculated the signal-to-noise ratio ($SNR = \mu/\sigma$) of expression levels across genes that carried the identified motifs (Fig. 4a and Supplementary Fig. 23). The spread of expression levels of motif-associated genes was over 2.5-fold larger than its expected (median) expression level (Supplementary Fig. 24), resulting in a low median *SNR* of under 0.4 (Fig. 4a). Of the full 4 order-of-magnitude range of gene expression levels observed in the data (Fig. 1a), only 57% could be recovered with the motifs, as indicated by the average expression level of genes per associated motif (Fig. 4b and Supplementary Fig. 23b). The range of gene expression levels for the identified motifs were thus too dispersed and overlapped, indicating that the predictions made by the model (Figs. 1f and 3b) were likely not based on single motif occurrences.

On the other hand, the observed interactions across the regulatory regions (Fig. 2a, b) and positional differences in expression levels (Fig. 3b) suggested that certain combinations of motifs found in different regulatory regions might carry a greater indication of expression levels than single motifs (Fig. 3f). We, therefore, searched for patterns of co-occurring motifs, i.e. combinations of motifs that are statistically more likely to be present together in genes than alone, and termed these patterns 'regulatory rules'. For this we used Market basket analysis—a technique that is commonly used for identifying frequently bought items in market research[75] ("Methods"). The number of identified rules corresponded to the quality of identified motifs and was largest at the 80% sequence identity cutoff, decreasing markedly when increasing the identity cutoff (Supplementary Table 11). A total of 9,962 rules were significantly over-represented (Chi-squared test[76] BH adj. *p*-value < 0.05) in at least 3 genes and across 93% of the analysed protein coding genome (Fig. 4c), and comprised 62% of all unique motifs that represented 86% of all motif occurrences (Supplementary Fig. 25).

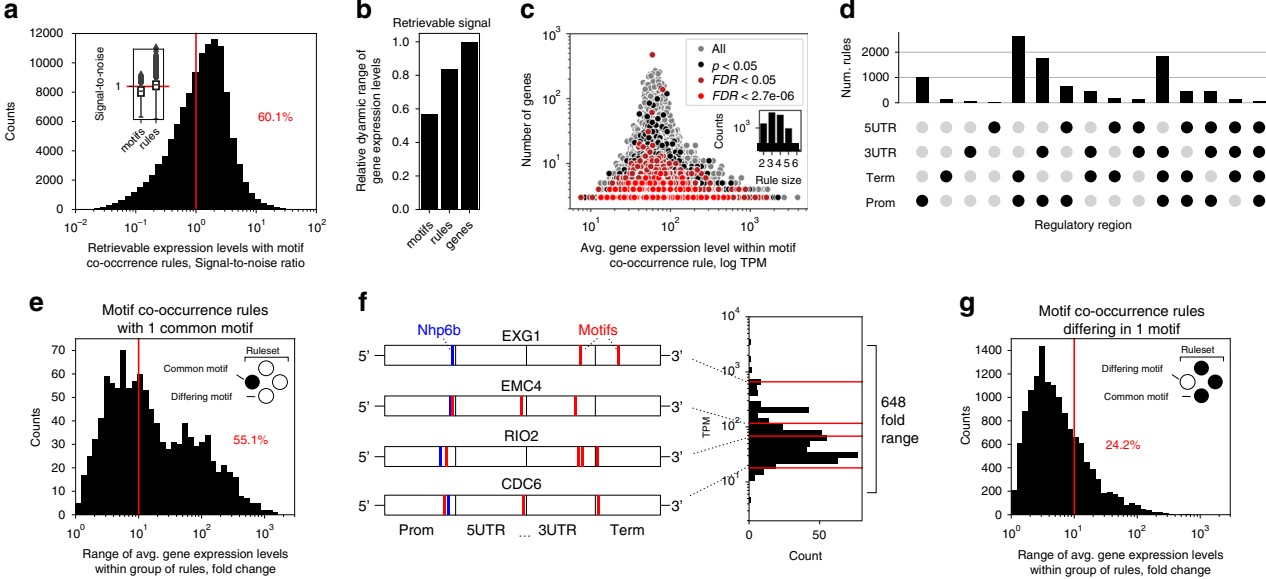

**Fig. 4 Motif co-occurrence uncovers the regulatory rules of gene expression. a** Distribution of the signal-to-noise ratio (*SNR*) of expression levels across genes carrying co-occurring motifs (co-occurrence rules, see Results text and Methods). Inset: comparison of *SNR* with single motifs ($n = 2098$) and with motif co-occurrence rules ($n = 116{,}734$). **b** The range of gene expression levels, relative to the full observed range in the initial RNAseq data (~4 orders of magnitude, Fig. 1a), that could be retrieved on average by genes carrying either single motifs or motif co-occurrence rules. **c** The amount of genes carrying a given motif co-occurrence rule versus the average expression level across the set of genes carrying the given rule, with increasing statistical significance levels from a chi-squared test[76]. *FDR* denotes Benjamini-Hochberg (BH) adjusted *p*-value and data is colored gray except for significance cutoffs $p < 0.05$ (black), FDR < 0.05 (dark red) and FDR < 2.7e-6 (equals Bonferroni correction, red). Inset: distribution of the number of co-occurring motifs in significant (FDR < 0.05) rules. **d** Distribution of motif co-occurrence rules across single or multiple *cis*-regulatory regions, according to the locations of the co-occurring motifs. **e** Distribution of gene expression levels with groups of motif co-occurrence rules that have one motif in common (i.e. unchanged). **f** Illustration of four genes (*CDC6, RIO2, NSP1, EXG1*) that carry a group of motif co-occurrence rules with a common motif (NHP6B transcription factor binding site, Tomtom[67] BH adj. *p*-value < 0.005, SGD:S000002157[104], blue line) in their promoter region, whereas they all diverge in possessing 2 to 4 other DNA motifs (red lines) across the remaining regulatory regions. These genes span a 648-fold range of expression levels. Red lines in the histogram denote the specific expression levels of the genes. **g** Distribution of gene expression levels with groups of motif co-occurrence rules that differ by a single motif. Source data are provided as a Source data file.

Importantly, the rules were frequently found in smaller groups of genes (at most 10 genes) and were crucial for discriminating genes with low and high expression levels, as the dynamic range of expression values decreased by 2-fold in rules present in more than 10 genes (Fig. 4c). Similarly, rules comprised of a larger number of motifs (Fig. 4c: rules of up to 6 motifs found) were also sparser and occurred in smaller groups of genes (Supplementary Fig. 26). In total, 88% of the significant rules (Chi-squared test[76] BH adj. *p*-value < 0.05) occurred across the regulatory regions compared to the rest that were present within single regions (Fig. 4d). This resulted in over 8-fold more co-occurring motifs spread across multiple regulatory regions than within any single region.

We next compared the genes carrying the single motifs to those carrying the co-occurring motifs in rules. We observed that the range of average expression levels of genes spanned by the rules exceeded those of single motifs by over 11-fold (Levene's test *p*-value < 1e-16) (Supplementary Fig. 24) and recovered, in total, 84% of the whole range of gene expression levels (Figs. 4b and 1a). Furthermore, a significantly (Wilcoxon rank-sum test *p*-value < 1e-16) narrower window of expression levels was observed with rule-associated genes compared to single motifs, as the variance of expression levels was over 16-fold lower (Supplementary Fig. 24). This showed that the genes containing co-occurring motifs fall under the precise control of the specific co-occurrence rule. The signal-to-noise ratio, which exceeded 1 for over 60% of rules, in contrast to single motifs, of which 78% were below 1 (Fig. 4a and Supplementary Fig. 23b), demonstrated that the precision of expression control of the rule-associated

genes was on average 3-fold higher (Wilcoxon rank-sum test *p*-value < 1e-16) than of single motifs (Supplementary Fig. 23). This again shows the presence of statistically measurable interactions across the entire gene regulatory structure, but this time at the level of motifs, thus supporting the existence of a coevolved interacting regulatory grammar.

To analyse how differences in the motif co-occurrence context across the regulatory regions affect the expression levels of genes, we grouped the motif co-occurrence rules, such that one motif was left unchanged and was common to all the rules, while the other motifs differed across the genes (Fig. 4e, rules with at least 3 motifs were used). We found that the motif co-occurrence context across the regulatory regions could repurpose a common motif in genes that exhibited a range of up to 1484-fold change of expression levels. Of the 1079 such rule sets that repurposed a given motif, 55% changed by at least one order of magnitude of expression levels (Fig. 4e). The repurposed motifs included significant (Tomtom[67] BH adj. *p*-value < 0.05) hits to a range of Jaspar TFBS in promoter regions, and co-occurred with as well as were repurposed by motifs across all the *cis*-regulatory regions (Supplementary Table 12). For example, one of the largest ranges of expression levels was observed with a group of NHP6B-like motifs (SGD:S000002157, 648-fold change), a TF that binds to and remodels nucleosomes. The NHP6B-like motifs co-occurred with motifs from the adjacent regulatory regions that govern the expression levels of many essential yeast genes (Fig. 4f), including those involved in DNA replication (*CDC6*), ribosomal RNA processing (*RIO2*), and coding for nuclear (*NSP1*) and cell membrane (*EXG1*) proteins. Furthermore, a similar trend was

observed by an alternative grouping of the motif co-occurrence rules, such that only one motif differed across the genes. Even a single motif substitution could achieve up to a 605-fold change of expression levels, where 24% of such rulesets were carried by genes that exhibited at least an order of magnitude change of expression levels (Fig. 4g). Finally, an analysis of gene sets containing the most widespread motif co-occurrence rules (that covered 20 or more genes) did not show any significant enrichment of specific cellular functions, which suggested that the uncovered gene expression grammar is shared across genes of all functions and is an integral part of the basic gene regulatory structure (Fig. 1d).

Since the regulatory regions were found to be coevolved with the coding region (Fig. 2e, f) and highly predictive of the codon frequencies (Fig. 2d), we analysed the properties of codon frequencies across the motifs and rules similarly as with the expression levels. The range of median Euclidean distances between codon frequencies, defined by the co-occurring motifs in rules, exceeded that of the single motifs by 6.5-fold (Levene's test $p$-value < 1e-16), whereas the average variance decreased almost 5-fold (Wilcoxon rank-sum test $p$-value < 1e-16, Supplementary Fig. 27). This shows that, compared to single motifs, co-occurrence rules define more conserved ranges of codon frequencies, and that the regulatory grammar learned by the deep models (and highly predictive of gene expression levels, Fig. 1f) is based on the combined properties of regulatory as well as coding regions.

**Regulatory properties learned from native genome guide expression engineering**. The common practice to manipulate gene expression levels is to use specific strong promoter (or terminator) sequences without much regard to which terminators (or promoters) they are combined with[44,45,77,78]. Considering that our findings point to a strong dependence of gene expression on the interaction between all regions of the gene regulatory structure (Figs. 2a, b, d and 4e, f, g), we used the deep neural networks (Fig. 1e, f) to explore how much the expression level of each gene can be altered with all the possible promoter-terminator combinations, a task that is challenging to perform experimentally. To rationally simplify this analysis and retain just two global halves of the gene regulatory structure, here 5′-UTRs were combined with promoters and 3′-UTRs with terminators (Fig. 5a). For each half, 17,960,644 combinations of the native yeast promoters or terminators with their corresponding variable counterparts were tested using the same dataset as for training the models (Fig. 1a: RSD < 1). We observed that, on average, varying the terminator region introduced a significant (Wilcoxon rank-sum test $p$-value < 1e-16) 3-fold change in either direction of expression levels, compared to the native gene structure (Fig. 5b). The terminator constructs achieved up to a 130-fold increase (Fig. 5b: YOL097W-A promoter with TDH3 terminator) and 14-fold decrease (Fig. 5b: TIM10 promoter with TOP3 terminator) of expression levels compared to the native counterparts. This showed that with a given gene, a range of over two orders of magnitude of expression levels (Fig. 5b: 130-fold range with YOL097W-A) could be unlocked merely by exchanging the terminator region. This held true for both strong as well as weak promoters (Fig. 5c). For instance, with the commonly used strong yeast promoter YEF3[77,78] we identified a regulatory combination where the expression levels decreased 3.2-fold (Wilcoxon rank-sum test $p$-value < 1e-16) compared to the native counterpart. Conversely, with the natively weak promoter POP6, a terminator context with a significant (Wilcoxon rank-sum test $p$-value < 1e-16) increase of over 7-fold was identified (Fig. 5c). A further comparison of natively weak and strong promoters (100 top and

bottom sorted constructs) confirmed that weak ones were expressed overall 2-fold more highly (Wilcoxon rank-sum test $p$-value < 1e-16), and strong ones overall 1.3-fold more lowly (Wilcoxon rank-sum test $p$-value < 1e-16) than the native sequences (Supplementary Fig. 28). The computed degree of regulatory freedom was supported also by published experimental results with the TDH3 promoter[45] (Fig. 5d). Here, despite the specific experimental conditions leading to an offset in the measurements, we observed a moderate correlation (Pearson's $r = 0.310$, $p$-value < 1e-16) to the variability of the fluorescence intensities with 4005 different terminators[45] as well as a comparable overall dynamic range (Fig. 5d).

When performing an analogous analysis using terminator regions and instead varying the promoters, expression levels changed on average over 20-fold (Wilcoxon rank-sum test $p$-value < 1e-16) in either direction and spanned a dynamic range of over 3 orders of magnitude (Fig. 5e: 2120-fold range with YNL146W). For all terminators, promoters could be identified that exerted a strong positive stabilizing effect, as the most pronounced variation was observed in the direction of increasing expression levels up to 1847-fold (Fig. 5e: YOL097W-A terminator with TDH3 promoter), compared with up to 32-fold decrease (Fig. 5e: MMF1 terminator with AMF1 promoter). Similarly as with promoters, strong and weak terminators (100 top and bottom sorted constructs) displayed stronger changes in the opposite directions of expression. Weak terminators were expressed overall 2.3-fold more highly (Wilcoxon rank-sum test $p$-value < 1e-16), and strong ones overall 2.4-fold more lowly (Wilcoxon rank-sum test $p$-value < 1e-16) than the native sequences (Supplementary Fig. 28), thus presenting many possibilities for gene expression engineering (Fig. 5f). Analysis of independent yeast experimental data[44] confirmed the computationally predicted variability with the ADH1 terminator, as we measured a strong correlation (Pearson's $r = 0.625$, $p$-value < 1e-16) to the variability of fluorescence intensities with the 625 tested promoters (Fig. 5g).

We considered that changing the promoters and terminators could have affected gene expression predictions based either on (i) the specific regulatory signals present in the DNA (Figs. 3f and 4h) or (ii) the general sequence properties, such as GC content and di-nucleotide composition[79,80]. We therefore evaluated the effect of removing high-order sequence information (ie. regulatory grammar) by randomly shuffling the regulatory DNA whilst preserving dinucleotide frequencies[81]. On average, the native DNA regions demonstrated a significant (Levene's test $p$-value < 1e-16) 88% larger effect on expression variation and a 13.5-fold higher dynamic range compared to randomly shuffled sequences with the same nucleotide composition (Supplementary Fig. 29). Accordingly, random sequences could only increase the expression level up to 4.9-fold (Supplementary Fig. 29: YIL102C-A promoter with a shuffled variant of its terminator) and decrease it up to 3.2-fold (Supplementary Fig. 29: NCE102 promoter with a shuffled variant of its terminator) compared to the native expression level. The observed increase of expression signal by combining native regions with their adjacent counterparts indicated that the correct progression of gene expression requires the presence of a specific regulatory grammar, similar to the one detected above (see Fig. 4h), which can be lost when artificially manipulating or carelessly combining the sequences.

The computational results suggest that the model-based predictions could improve experimental design of gene expression systems and potentially decrease a part of the observed variability of gene expression levels (Fig. 5b, e). Therefore, using deep models to guide the development of experimental constructs, we set out to experimentally verify that a weak promoter can be driven to higher expression levels and vice versa,

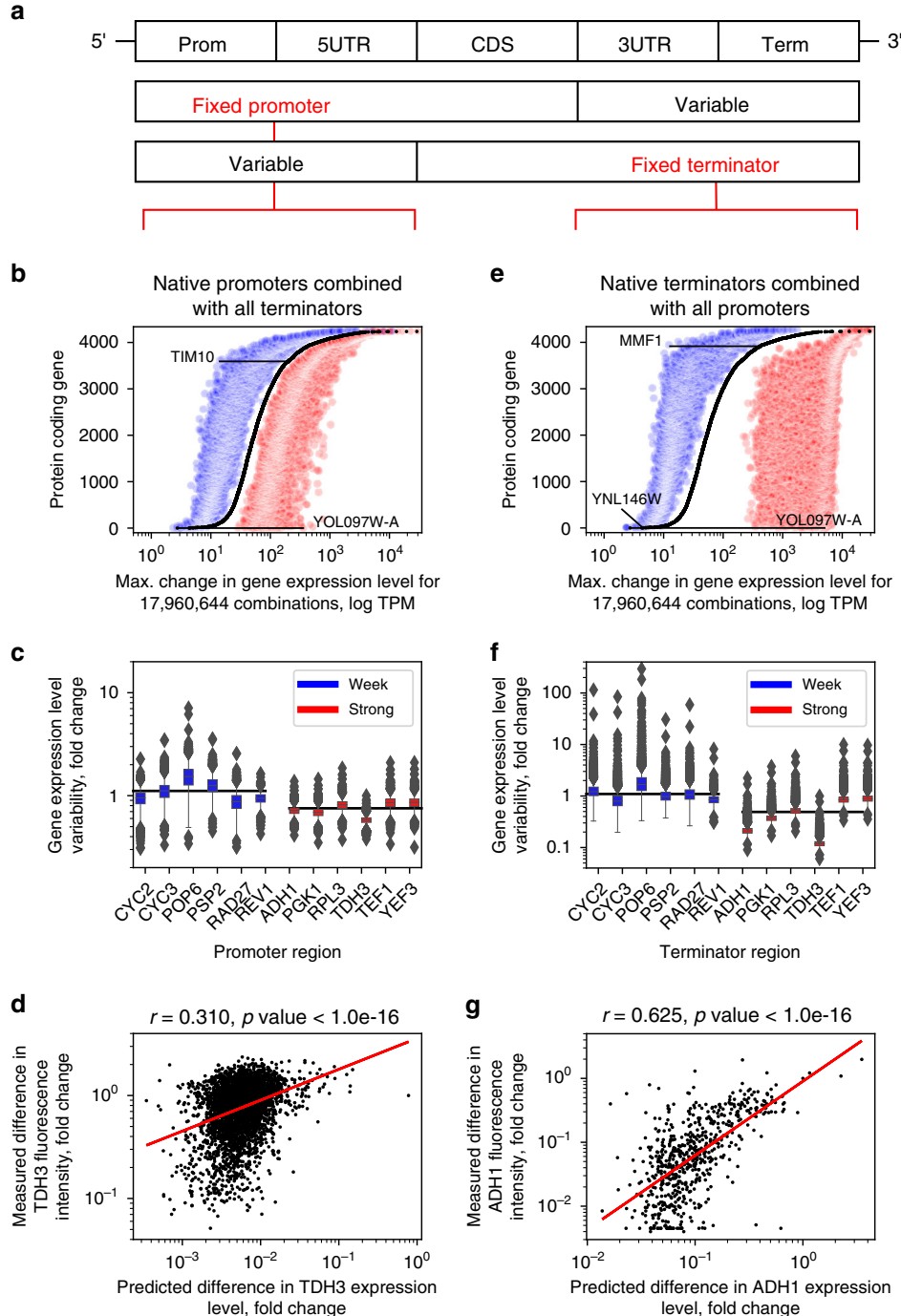

**Fig. 5 Each gene is predicted to contain a large degree of regulatory freedom for expression engineering. a** Two global halves of the gene regulatory structure where 5′-UTRs were combined with promoters and 3′-UTRs with terminators. **b** Maximum increases (red) and decreases (blue) of gene expression levels with combinations of 4238 native promoters with all terminator variants. **c** Distributions of gene expression levels for combinations of either strong (red) or weak (blue) promoters with all terminator variants ($n = 4238$ for each construct). Black lines denote median exp. levels. **d** Comparison of changes in predicted *TDH3* expression levels with measured fluorescence intensities when combining the promoter with $n = 4005$ terminator variants[45]. Red line denotes least squares fit. **e** Maximum increases (red) and decreases (blue) of gene expression levels with combinations of 4238 native terminators with all promoter variants. **f** Distributions of gene expression levels for combinations of either strong (red) or weak (blue) terminators with all promoter variants ($n = 4238$ for each construct). Black lines denote median expression levels. **g** Comparison of changes in predicted *ADH1* expression levels with measured fluorescence intensities when combining the terminator with $n = 625$ promoter variants[44]. Red line denotes least squares fit. For box plots in **c**, **f**, boxes denote interquartile (IQR) ranges, centres mark medians and whiskers extend to 1.5 IQR from the quartiles. Source data are provided as a Source data file.

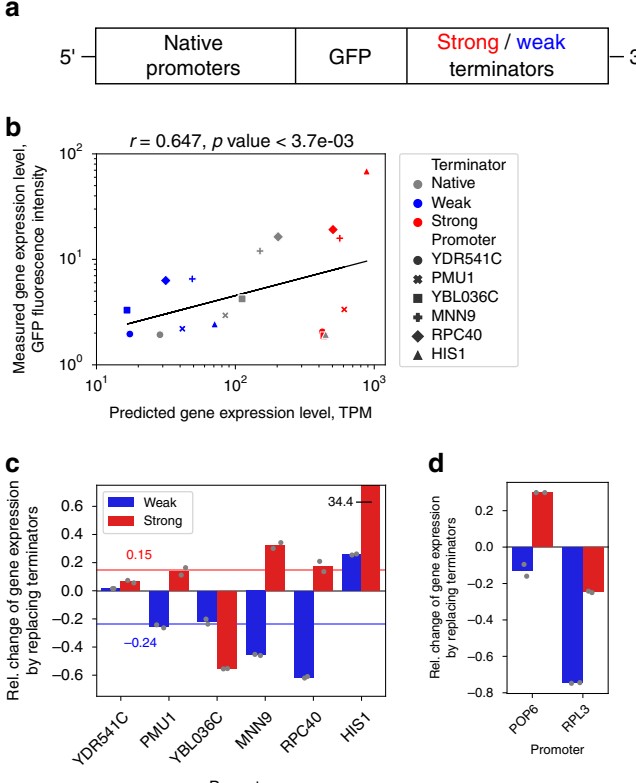

**Fig. 6 Experimental analysis demonstrates the potential of using the models to fine-tune gene expression.** **a** Conceptual diagram of experiment and constructs. **b** Correlation analysis between model predictions and measured GFP fluorescence levels for constructs with 6 promoters in combination with native (grey) as well as strong (red) and weak (blue) terminators ($n = 18$). Black line denotes least squares fit. **c** Relative change in measured GFP fluorescence levels of the constructs, where native terminators were replaced with either strong (red bars) or weak (blue bars) variants with $n = 2$ technical replicates indicated as grey points (see text and Supplementary Table 13). On average, the GFP fluorescence levels shifted by 20% in the directions of the predicted levels. Red and blue lines denote median values with strong and weak promoters, respectively. **d** Additional tests with the weak POP6 and strong RPL3 promoters, whose CDS were more dissimilar to GFP. Native terminators were replaced with either strong (red bars) or weak (blue bars) variants with $n = 2$ technical replicates indicated as grey points. Source data are provided as a Source data file.

based on substituting its terminator with another natural variant (Fig. 6a, "Methods"). Since here the experimental design required models with limited 500 bp regions on either side of the CDS, we constructed a new model based on constricted input data that contained only the most relevant parts of the regulatory regions, as determined from the relevance profiles (Fig. 3a: 400 bp, 100 bp, 250 bp and 250 bp, respectively, Supplementary Fig. 30a). This model achieved similar performance ($R^2_{test} = 0.797$, Supplementary Fig. 30b and 31) as the previous one, despite requiring regulatory sequences that were approximately half the length. Compared to the above computational analysis, where only one side of the regulatory regions was perturbed (Fig. 5a), development of experimental constructs for the fluorescence measurements additionally required the substitution of all coding sequences (codon frequencies) with a GFP variant (Methods), which represented approximately half of the model input response (Supplementary Fig. 31). In order to retain the predictive power of the model despite these large perturbations,

we selected 6 promoter variants from the set of constitutive genes. The data was subset to the 10th percentile, based on the similarity between native and GFP coding sequences, as well as the model predictive accuracy (Supplementary Fig. 32). These GFP-substituted genes covered a 16-fold range of predicted expression levels from 26 to 420 TPM (Supplementary Table 13). Similarly, a weak and a strong terminator were selected according to the strength of their effects on increasing or decreasing the GFP-substituted gene expression levels compared to the native terminators (Supplementary Table 13). Experimental measurements with the resulting 18 constructs (Fig. 6a: 6 each with native terminator, weak terminator, strong terminator, respectively, Methods) achieved good correlation to the predicted levels (Pearson's $r = 0.647$, $p$-value $< 1.8e-3$, Fig. 6b). We observed, on average, a corresponding 20% change in GFP fluorescence levels with all constructs, in accordance with the predicted expression level changes (Supplementary Table 13), with the exception of the YBL036C promoter with the strong terminator and the HIS1 promoter with the weak terminator (Fig. 6c). We identified a strong promoter with HIS1 that consistently (duplicate tests) achieved a 3440% increase in GFP intensity (Supplementary Fig. 33) and, conversely, a weak promoter with RPC40 leading to a 61% decrease in GFP intensity (Fig. 6c). In addition, we tested the well known weak POP6 and strong RPL3 promoters[77,78], which showed considerable divergence of properties and model predictions between the native and GFP coding sequence (Supplementary Table 13 and Supplementary Fig. 34). Despite this, we were able to obtain a weak terminator variant with the RPL3 promoter leading to strong decrease in gene expression (Fig. 6d: 75% decrease in GFP intensity), whereas the POP6 promoter achieved a smaller 30% increase and 13% decrease, respectively. To conclude, even in the highly perturbed state with multiple substitutions to the gene regulatory structure, our models enabled computational pre-screening of optimized candidates from millions of combinations and thus demonstrated the feasibility of model guided fine-tuning of gene expression levels.

## Discussion

In the present study we asked the question: to what extent are gene expression levels encoded in the coding and *cis*-regulatory DNA regions of the gene regulatory structure? This question followed our observation that with all known biological variation of gene expression, based on a collection of RNA-Seq experiments across the majority of presently tested conditions, 79% of protein coding genes were merely within a ±1-fold change of their median expression levels in 2/3 of the conducted experiments (Fig. 1a). By training deep neural networks on the sequences from the entire gene regulatory structure (Fig. 1d), we demonstrated that most gene expression levels in *S. cerevisiae* are predictable using only DNA encoded information (Fig. 1f: $R^2_{test} = 0.822$). Therefore, 4 orders of magnitude of the transcriptional repertoire can be directly determined from the DNA sequence, irrespective of the experimental condition. Of course, this statement does not object to the reality that regulation of gene expression can be highly dynamic between different conditions[9,44]. However, for the majority of genes, this biological variation across conditions is much smaller than the magnitude of expression levels between the genes (Fig. 1a), meaning that genes which are highly expressed in the majority of conditions will likely always (79% of protein coding genes) be highly expressed and vice-versa. These differences in gene expression levels are encoded in the DNA and can be read using machine learning. Accordingly, similar results were obtained in 6 other model organisms spanning multiple kingdoms of life (including bacteria and eukaryotes such as fruit

fly, zebrafish, mouse and human), that covered the complete range of genetic regulatory complexity[46], as measured by coding gene density (Supplementary Table 1).

The models in the present study mapped the relations of input DNA sequence variables to the continuous target mRNA levels, where model performance was assessed with the coefficient of determination $R2$ on the test data. This is an intuitive measure[12,31,34] signifying the proportion of the variance in the dependent variable (mRNA levels) that can be predicted from independent DNA sequence variables (Fig. 1d and Supplementary Fig. 1c)—i.e. how much could be learned from the sequence data using our machine learning approach[82,83]. We emphasize that for the actual model selection during training, the mean squared error (MSE) was used, as this is a robust goodness-of-fit measure that makes no assumptions about the data distributions[84,85]. For each model: (i) the model was trained on the training set using minimization of MSE criteria, (ii) Hyperparameter tuning was performed on the validation set, (iii) a model with minimal MSE on the validation set was chosen and (iv) for final testing evaluation the model with the minimal MSE criteria was accessed again to ensure that it did not overfit the data (Supplementary Tables 3 and 8). As an alternative statistical approach and a measure of goodness of fit, we also performed the $F$-test with all models (Supplementary Tables 3 and 8). Although not informative about a model's predictive performance, this tested whether the model in question fits the data significantly better than the null model based only on the intercept (i.e. average over data) to make predictions[85]. Moreover, to clearly differentiate between reporting model performance with $R^2$ and the strength of a linear relation between independent variables (such as predicted expression levels and measured GFP fluorescence in Fig. 1g), for the latter the Pearson's correlation coefficient $r$ was used.

Since individual coding and non-coding parts of the gene regulatory structure (Fig. 1d) all play a crucial role in the regulation of gene expression[9], we inquired which particular regions contained the highest amounts of information on gene expression levels. While codon frequencies were highly informative about mRNA levels (Fig. 2b), we observed that a similar amount of information was encoded in the flanking regions (Fig. 2a). Indeed, this was further supported by the result that deep neural networks could predict the codon usage of a gene merely based on that gene's regulatory sequence (Fig. 2c, d). Although the effect of codon usage on overall transcription has been widely studied and debated[86–88], their surprisingly strong effects on transcription, also supported by our results (Fig. 2b), have only recently started coming to light. The hypothesized mechanisms through which codons affect transcription are: (i) effects on nucleosome positioning[38], (ii) premature termination of transcription, usually by mimicking poly-A signals[89], or (iii) mRNA toxicity[90]. Generally, the differences in codon usage between bacterial species can be explained from the dinucleotide content in their non-coding regions[91] and it is widely assumed that most interspecies variation in codon usage is attributed to mutational mechanisms[86,92]. Within a genome, however, given that codon usage can be predicted from non-coding regions (Fig. 2d), and both coding and regulatory regions are similarly predictive of gene expression levels (Fig. 2b), it is likely that the entire gene regulatory structure undergoes a common selection pressure. Multiple lines of evidence support this both in higher eukaryotes[62,63,93–97] and yeast[59,98–100], as (i) approximately half of all functional variation is found in non-coding regions[62], with *cis*-regulatory regions undergoing not only purifying negative[97] but also positive selection[63,93,94,98,101], (ii) findings indicate a similar selective pressure on gene expression and protein evolution[95,96,99], and (iii) instances of coupled protein and regulatory evolution were observed[59,60,95]. Accordingly, mutation

rates in orthologs from 14 yeast species supported the notion of coevolution between the coding and regulatory regions (Fig. 2e, f), indicating that the gene regulatory structure is a co-evolving unit. However, despite the ensuing information overlap among the different regions (Fig. 2b, d: up to 58%), each region contributed to mRNA level prediction (Fig. 2a, b) and the entire gene regulatory structure was required to define over 82% of the gene expression variability (Fig. 1f).

The multiple regulatory elements of the gene regulatory structure individually or jointly control the different DNA processing phases required for mRNA transcription, including nucleosome positioning, mRNA synthesis, and mRNA maturation and decay[9] (Supplementary Fig. 1a). To control enzyme interactions, DNA, therefore, contains a plethora of statistically identifiable DNA motifs[68,69]. The question remains though, which of those motifs are relevant signals for regulating mRNA levels? For instance, based on currently identified JASPAR motifs[68], a significant enrichment of known promoter TF binding sites was found particularly in highly expressed genes (Supplementary Fig. 21). However, such an analysis does not uncover which motifs or combinations of motifs is found to be important by a predictive model of mRNA abundance (Fig. 1e). To resolve this, we opened the neural network "black box" (Fig. 1e and Supplementary Fig. 14) and determined which DNA sequences were causing significant neural network responses (Fig. 3a). Although thousands of DNA motifs were uncovered across all regulatory regions (Fig. 3d, f), individual motifs could not explain the entire dynamic range of gene expression, i.e. the same motifs were found in both lowly and highly expressed genes (Fig. 4a, Supplementary Fig. 23). However, by statistically analysing the interactions between motifs, we could retrieve a much more accurate (Fig. 4a) and comprehensive (Fig. 4b) indication of expression levels than with single motifs, where an almost 3-fold higher amount of co-occurrence rules surpassed a SNR of 1 (Fig. 4a) and recovered almost the whole (84%) dynamic range of expression levels as opposed to 57% with motifs (Fig. 4b). 9,962 combinations of 2 to 6 motifs were found to co-occur more frequently together than alone across the gene regions (Fig. 4c) and were informative of almost the entire dynamic range of expression (Fig. 4a, b, f). Moreover, the motif co-occurrence rules also defined more specific ranges of codon usage than single motifs (Supplementary Fig. 27), further supporting our results that the entire gene regulatory structure (Fig. 1d), including both coding and non-coding regions, is a single co-evolved interacting unit (Fig. 2d, e, f).

Finally, we demonstrated that deep neural networks can learn the complex regulatory grammar of gene expression directly from an organism's genome (Fig. 1f, h), without any prior knowledge of genetic regulation or the need to perform laborious high-throughput screening experiments with synthetic constructs. Despite the fact that the machine learning model had never seen the synthetic DNA data, it was able to successfully recapitulate fluorescence readouts from published experimental studies[44,45] and demonstrate a strong agreement between model predictions and experimental measurements (Fig. 1g and Supplementary Fig. 5), even on constructs containing de novo generated sequences[22] (Supplementary Fig. 5c). Furthermore, since the trained models encapsulate the whole interacting regulatory grammar that must be present to correctly drive expression, they can be used to enhance experimental techniques and improve control over gene expression in synthetic biology. We show that the standard approach of introducing a variety of terminators (or promoters) in combination with strong promoters (or terminators) can in fact lead to large variability in each direction of the actual measured levels of expression (Fig. 5c, f)[45,77]. Our model was however ~1.6-fold more capable of predicting

downregulation compared to upregulation (Fig. 6b), which was likely due to the average native expression level being over 2-fold higher from a predicted average basal expression level (Supplementary Fig. 35a: 64.5 TPM). From an evolutionary perspective, this suggests that the regulatory grammar might be specifically adapted not only for higher expression levels but also for lower ones below the basal expression level, whereas around the basal level the grammar is less specific and possibly more diverse (Supplementary Fig. 35c). This is also in agreement with the observations that highly expressed genes are under stronger selection pressure than average ones[102] and thus altering these highly optimized sequences likely results in downregulation. The implications for future experiments are that (i) separate models for different classes of expression as well as accounting for different conditions or tissues to decrease the experimental variation might provide more accurate predictions, and (ii) more computational and experimental work is required to decipher the evolutionary strategies of regulatory grammars and define the properties of underlying basal and targeted-evolved regulation. Nevertheless, using the deep learning models, researchers can now computationally test the effects of the different combinations of regions (Fig. 5b, e), even with specific perturbations to gene coding sequences (Fig. 6b). This way, they can obtain candidate variants that enable rapid development of constructs with desired levels of gene expression (Fig. 6c, d). This has potential to greatly decrease experimental noise, accelerate experimental throughput and thus decrease the overall costs of microorganism and cell-line development in biotechnology[3].

## Methods

**Data**. Genomic data, including gene sequences, as well as transcript and open reading frame (ORF) boundaries, were obtained from Ensembl (https://www.ensembl.org/index.html)[103]. The exceptions were organisms (i) *S. cerevisiae* S288C, for which data were obtained from the Saccharomyces Genome Database (https://www.yeastgenome.org/)[104,105] and additional published transcript and ORF boundaries were used with this organism[106,107], and (ii) *E. coli* K-12 MG1655, where all data were obtained from the RegulonDB database (http://regulondb.ccg.unam.mx/)[108] (Supplementary Table 1). For each organism, coding and regulatory regions were extracted based on the transcript and ORF boundaries. DNA sequences were one-hot encoded, UTR sequences were zero-padded up to the specified lengths (Fig. 1d and Supplementary Figs. 9 and 10), codon frequencies were normalized to probabilities, and 8 mRNA stability variables were computed that included: lengths of 5′-UTR, ORF and 3′-UTR regions, GC content of 5′- and 3′-UTR regions, and GC content at each codon position in the ORF.

For gene expression levels, processed raw RNA sequencing Star counts were obtained from the Digital Expression Explorer V2 database (http://dee2.io/index.html)[32] and filtered for experiments that passed quality control. Raw mRNA data were transformed to transcripts per million (TPM) counts[109] and genes with zero mRNA output (TPM < 5) were removed (Supplementary Table 2). Prior to modeling (Supplementary Fig. 7), the mRNA counts were Box-Cox transformed[110] (see lambda parameters in Supplementary Table 4).

Modeling datasets were obtained with the above data processing and comprised paired gene regulatory structure explanatory variables (Fig. 1d and Supplementary Fig. 1c) and mRNA abundance response variables (Fig. 1a). No significant pairwise correlations were found between the variables (Supplementary Fig. 4 and Supplementary Table 5), except between mRNA counts and ORF lengths, due to the technical normalization bias from fragment-based transcript abundance estimation[111]. To obtain mRNA counts that were uncorrelated to gene length, the residual of a linear model, based on ORF lengths as the response variable and mRNA counts as the explanatory variable, was used as the corrected response variable (Supplementary Fig. 4). When testing whether the introduced correction could potentially remove biological signal associated with gene length, using data from whole molecule RNA sequencing that does not rely on short-read assembly[112], no correlation between gene length and its expression was found (Pearson's $r = -0.08$, $p$-value < 1e-6; Supplementary Fig. 8).

**Statistical hypothesis testing**. For enrichment analysis, gene ontology slim terms[113,114] were obtained from the Saccharomyces Genome Database[104,105] and published promoter classifications were used[27,65]. For statistical hypothesis testing, Scipy v1.1.0 was used with default settings. All tests were two-tailed except where stated otherwise.

**Supervised deep methods**. Different neural network architectures were tested that combined: (i) 1 to 4 convolutional neural network (CNN) layers[115] (see tested parameters in Supplementary Table 6), which included inception layers[116] (ii) 1 to 2 bidirectional recurrent neural network (RNN) layers[117], and (iii) 1 to 2 fully connected (FC) layers, in a global architecture layout CNN-RNN-FC[31,41,118,119]. Training the networks both (i) concurrently or (ii) consecutively, by weight transfer on different variables (regulatory sequences to CNN and RNN, numeric variables to FC), showed that the architecture yielding best results was a concurrently trained CNN (3 layers)-FC (2 layers)[12,40,120,121], which was used for all models. Batch normalization[122] and weight dropout[123] were applied after all layers and maxpooling[124] after CNN layers (Supplementary Table 6). The Adam optimizer[125] (Supplementary Table 6) with mean squared error (MSE) loss function and ReLU activation function[126] with uniform[127] weight initialization were used. In total, 26 hyper-parameters were optimized using a tree-structured Parzen estimators approach via Hyperopt v0.1.1[128] at default settings for 1500 iterations[129,130].

The explanatory input data and corresponding response variables were divided into training (80%), validation (10%) and test (10%) sets. Tests with different sizes of input regulatory sequences showed that whole regions resulted in the most accurate models (Supplementary Fig. 10). The best models were chosen according to the minimal MSE on the validation set with the least spread between training and validation sets. The coefficient of determination ($R^2$) computed on the test set is reported in the main text and was defined as $R^2 = 1 - SS_{Residual}/SS_{Total}$ [Eq. 1], where $SS_{Residual}$ is the sum of residual squares of predictions and $SS_{Total}$ is the total sum of squares, and statistical significance was evaluated using the two-tailed $F$-test.

To assess model predictions by varying either promoter[44] or terminator[45] regions, input explanatory variables were constructed based on the specified coding (fluorescence reporters codon frequencies) and regulatory regions (variable region combined with specified adjacent regions). For building and training models Keras v2.2 and Tensorflow v1.10 software packages were used and accessed using the python interface.

**Supervised shallow methods**. For shallow modeling the following regression algorithms were used: linear regression, ridge regression, lasso, elastic net, random forest, support vector machines with nested cross-validation, and k-nearest neighbour regression[131]. To include information from the regulatory DNA sequences in the shallow models, k-mers of lengths 4–6 bp were extracted from the regulatory DNA sequences[55–58] and used as additional explanatory variables. Nested cross validations by selecting the best models with the lowest MSE on held-out sets were performed with these algorithms and GridSearchCV using the Scikit–learn package v0.20.3 with default settings. The coefficient of determination ($R^2$) (Eq. 1) computed on the test is reported in the main text analogously as with deep models in M3.

**Analysis of evolutionary rates**. Multi-sequence alignments of 3800 orthologous protein-coding genes (each gene divided into separate promotor, 5′- and 3′-UTR, terminator and coding regions) from fourteen fungal species were generated using Mafft v7.407[132] with the Linsi algorithm and default settings. Orthologs were defined according to Ensembl Compara[103]. The resulting 19,000 alignments were analysed with Zorro v1.0[133] to identify regions of high sequence variability and possible misalignment that could have a negative effect on the phylogenetic signal in the overall sequence. After excluding sites with a confidence score ≤ 0.2, each individual alignment was analysed for 1,000,000 generations in MrBayes v3.2.6[134], with the number of substitution types set to one, and a gamma distribution of substitution rates, to obtain the estimated mean substitution rate (alpha) for each dataset.

**DNA relevance analysis**. To calculate the relevance of different DNA sequences for model predictions, defined as Relevance $= (Y - Y_{Occluded})/Y$ [Eq. 2], where $Y$ is the model prediction, an input dataset with sliding window occlusions was used with the deep models to obtain predictions[64,135] (Supplementary Fig. 14). The window size of the occlusions was set to either: (i) whole regions, to analyze the *relevance* of region combinations and sensitivity analysis or (ii) 10 bp for motif analysis, determined based on the analysis of *relevance* profiles at difference occlusion sizes using the FastDTW v0.3.2 alignment method[136] and analysis of the distribution of DNA sequence motif sizes in the JASPAR database[68] (Supplementary Fig. 15). For clustering of *relevance* profiles, the consensus clustering approach[137] was used, as implemented in the package ConsensusClusteringPlus v1.48.0 with the method Partition around medoids (pam) set to 50-fold subsampling of 80% of data points and using the Pearson correlation distance. The number of clusters ($k$) was determined at $k = 4$, for which the relative consensus did not increase more than 10% (Supplementary Fig. 22).

**DNA motif analysis**. Extraction of relevant sequences, i.e. those that significantly (standard deviation ≥ 2) affected gene expression prediction, yielded 169,763 sequences that spanned all the analysed genes ($RSD < 1$) (Supplementary Fig. 18). To identify regulatory motifs, clustering of the relevant sequences was performed using CD-HIT v4.8.1[138,139], with recommended settings (a $k$-mer size of 4, 5 and 6 was used in correspondence with sequence identity cutoff 0.8, 0.85, and 0.9, respectively) and a cluster size of 5 sequences, based on the amount of

recovered sequences and unique motifs. Multiple sequence alignment on the clustered sequences was performed using Mafft v7.407[132], with the Linsi algorithm and default settings. Position weight motifs (PWMs) were processed using the Biopython package v1.73[140] and motif edges were trimmed below a cutoff of 0.2 bits[141]. Pairwise comparisons of the PWMs across the regulatory regions and comparisons to JASPAR[68] (core fungi, non-redundant) and Yeastract[69] databases was performed using Tomtom and Meme suite v4.12[67,142], with recommended settings. Motif co-occurrence was analysed using the FP-growth algorithm as implemented in Apache Spark v2.4[143], with default settings accessed through the Python interface, where *support*, *confidence* and *lift* were calculated as defined in[75]. The statistical significance level of co-occurring motifs was determined using the chi-squared test[76].

**Experimental strain construction**. The *S. cerevisiae* strain CEN.PK113–7D[144] was used as the base strain for all genetic engineering. Integration of the promoter-GFP-terminator constructs was done using the CRISPr/Cas9 plasmid as well as the gRNA Helper vectors from the EasyClone marker-free system at the XI-2 locus (using pCFB2312 + pCFB3044)[145]. All transformation steps were performed according to the published manual, with the exception that the repair fragment was provided as three fragments: the promoter with 30 bp overlap to the genome and 30 bp overlap to the GFP gene, the GFP gene, and the terminator with 30 bp overlap to the GFP gene and 30 bp overlap to the genome. For each fragment, 250 µg of DNA were used for the transformation. The DNA fragments for the promoters and terminators were obtained using PCR (see Supplementary Table 14 for the list of primers). All PCR products were purified using the Thermo Scientific GeneJET PCR Purification Kit. For the GFP gene, the *UBIMΔkGFP** version from Houser et al.[146] was used (see Supplementary Table 15 for sequence), where the DNA sequence was ordered as a gene fragment from Eurofins.

**Fluorescence measurements and analysis**. The yeast cells were pre-cultured overnight in 96 deep well plates in 0.5 ml of minimal media with 2% glucose (see Supplementary Table 16 for media composition) at 30°C and 300 rpm. The following day, the cultures were set up in 96 deep well plates with a starting OD600 of 0.1. After 5 h of cultivation, when the cells were in mid-exponential growth phase, the cells were diluted with water to a final OD600 of 0.02 in a total volume of 200 µl in 96-well round plates in technical duplicates. Using the Guava easyCyte 8HT flow cytometry system the GFP (green fluorescence) intensity as well as cell size (forward scatter) and granularity (side scatter) were measured (Supplementary Fig. 36). The cells were gated based on the forward and side scatter intensity in order to exclude a few very large and very small cells (Supplementary Fig. 37). For each technical duplicate the median GFP intensity value is reported.

**Software**. Python v3.6 (www.python.org) and R v3.6 (www.r-project.org) were used for computations. Code for the data analysis is available at https://github.com/JanZrimec/DeepExpression[147] and data at https://doi.org/10.5281/zenodo.3905251 (see also section on Data Availability).

**Reporting summary**. Further information on research design is available in the Nature Research Reporting Summary linked to this article.

## Data availability

Genomic data, transcript and gene boundaries were obtained from Ensembl Genomes release 41 and Ensembl release 94 (https://www.ensembl.org/), Saccharomyces Genome Database (https://www.yeastgenome.org/) and RegulonDB v10.5 database (http://regulondb.ccg.unam.mx/) (links to raw data in Supplementary Tables 4 and 10). RNA sequencing data was obtained from the Digital Expression Explorer V2 database (http://dee2.io/mx/), DNA sequence motifs from the Meme suite motifs databases file (http://meme-suite.org/) and additional data from the cited references (links to raw data in Supplementary Table 7). The Source Data file was deposited to the Zenodo repository and is available at https://doi.org/10.5281/zenodo.3905251.

## Code availability

Code for the data analysis was deposited to the Github repository and is available at https://github.com/JanZrimec/DeepExpression[147].

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

## Acknowledgements

We thank Hal Alper for kindly sharing data from his lab, Victor Garcia for insightful discussions on codon usage, and Clara Correia-Melo and Simran Aulakh for commenting on the manuscript. We gratefully acknowledge the NVIDIA Corporation for supporting this research as well as the Chalmers Centre for Computational Science and Engineering (C3SE) and the Swedish National Infrastructure for Computing (SNIC) for providing computational resources. Mikael Öhman and Thomas Svedberg at C3SE are acknowledged for technical assistance. J.Z., F.B. and A.Z. were supported by SciLifeLab funding. CSB was supported by the European Union's Horizon 2020 research and innovation programme [Marie Skłodowska-Curie grant agreement No 722 287]. The study was supported by the BIGDATA@Chalmers funding initiative (Area of Advance ICT).

## Author contributions

J.Z. and A.Z. conceptualized the project; J.Z., F.B., A.S.M., R.C., V.V., M.T. and A.Z. designed and performed the computational analysis; J.Z., C.B., V.S. and A.Z. designed the experimental analysis; C.B. performed the experimental analysis; J.Z. and A.Z. interpreted the results; J.Z. and A.Z. wrote the initial draft paper; J.Z., C.B., F.B., V.S., J.N., M.T. and A.Z. revised the initial draft and wrote the final paper.

## Funding

## Competing interests
The authors declare no competing interests.
