## [Peer Review File · Nature Communications]

Reviewers' comments:

Reviewer #1 (Remarks to the Author):

The authors present a very information dense, machine learning based approach to understand gene expression from DNA sequence ultimately achieving modest R² values across a set of model organisms from bacteria to humans using a deep neural network. While the work is potentially of interest, the overall message and ultimate utility is lost in the delivery of the paper that was very dense to read and understand. Specifically:

The use of R² as a metric is not the best to define a model and it would be suggested to use alternative statistics approaches to show a goodness of fit such as an F-score as an example. In some cases, the goodness of fit does not seem very strong despite high R² (take for example Figure 6B).

A comparison of this machine learning approach to other approaches (including things like the RBS calculator and other predictive approaches) should be used to determine the appropriateness of this learning method and the resulting models.

The predictive power of the approach to a de novo sequence would be suggested.

The authors set boundaries for the various promoter, terminator etc elements that (in some organisms) would overlap with neighboring genes. How often was this the case? How much influence does this exact window have on the determination of function from sequence? How much would a promoter (as an example) taken out of context be influenced by the genomic context? Can this be predicted by the model?

Extracted motifs such as those in Figure 3 should be head-to-head compared to other work in the literature that have extracted similar phenomena.

The paper needs to be completely revised for clarity of reading.

Reviewer #2 (Remarks to the Author):

The authors propose a very interesting computational method to try and deduce the DNA regulatory code underlying gene expression changes in several organisms. They are able to explain a high percentage of the gene expression changes using a deep learning approach and look into the effect of (combinations of) different parts the regulatory elements on gene expression.

While I think this is a very interesting paper with important findings of interest to a wide audience, I do think that the paper needs more attention, in particular with regard to overall readability and also addressing/mentioning the drawbacks.

Major points:

- I found the first part of the results (pg 6, 7 model definition) a bit hard to get the exact details of how the model was obtained. At various times, the authors refer to the methods, but was unable to unambiguously find the section in the methods section. It would be really helpful if the authors provide a more extensive methods (or maybe supplemental methods) section with regard to the data processing (maybe split it up by data type?) and parameters used for fitting. This was now an exercise in trying to match the main text with the methods and supplemental figures, without a clear guidance.

- Throughout the text it remains a bit unclear when the authors switch from the analyses

performed for all seven organisms to the indepth analyses performed using yeast data. It would help alot if the authors make much clearer that the majority of the results and conclusions are based on yeast. Only the global variance estimates where verified in other organisms, which also clearly showed that more complex organisms are harder to predict.

- I am not convinced by the co-evolution analyses. While I'm not an evolutionary biologists, claiming that regulatory regions co-evolve with the corresponding coding regions based on just one correlation analysis seems a bit overenthusiastic. How does for instance the correlation look like with other parts of the genome that are not directly related, is that a much lower number?

- The authors seem to ignore the fact that their model is much more capable of predicting downregulation compared to upregulation (pgs 7 and 18). This warrants some thoughts in the discussion about the implication of their model predictions for future experiments and a potential explanation why this might happen (can imagine that down more often equates to off and therefore is easier to predict than up).

- I am intrigued by the difference they found between the shallow and deep modeling with regard to the contribution of the regulatory regions. While the shallow modeling is unable to explain a large part of the variance, the deep modeling is. When combined with the codon frequencies, the difference between the two approaches is much smaller. The authors also indicate that they can predict the codon usage using the regulatory code. How then can such a big difference between the two modeling approaches exist? The shallow modeling suggests that the codon frequency is much more important compared to the regulatory regions, while the deep modeling is assigning a much higher contribution to the regulatory regions.

minor remarks:

- pg 6, FigS1-2, in contrast to what they authors indicate, I only see metabolic processes, transport and stress response in the enrichment analysis.

- Fig1G, Why use a pearson r here, while in all other subpanels an R2 is used. This is confusing.

- pg 9, the authors claim that the combination is additive "Each DNA region thus additively contributed ..." This warrant some further explanation. Effects can also be non-additive (multiplicative for instance), why/how did the authors conclude this?

- pg 22, discussion. There, the authors indicate that the biological variation due to different conditions is negligible compared to the gene expression difference encoded in the genome. I think the authors have to be a bit more careful here with their statement. I would disagree that biological variation is negligible. In essence, the parameters they choose to include in their model (motifs, regulatory sequence) are effectively ways for the system to encode responses to different environmental changes through transcription factor binding, chromatin rearrangements, etc and therefore should be considered an indirect way of environmental changes on gene expression levels.

We would like to thank the reviewers for their thoughtful comments and efforts towards improving our manuscript. We address specific comments to each reviewer below. Reviewers had very different comments, but the general overall concern was about manuscript clarity. While we addressed the specific concerns and provided corresponding additional analyses, we also considerably rewritten the manuscript.

Many of the additions aimed at addressing the manuscript clarity; we have added multiple paragraphs, expanded Methods section, shortened the sentences, introduced better transition sentences, and expanded the discussion section. The manuscript was also proofread by 2 native English speakers.

In the following pages, we respond to the comments that were more specific to each referee.

Referee 1

Comment 1.1a

Reviewers' comments:

Reviewer #1 (Remarks to the Author):

The authors present a very information dense, machine learning based approach to understand gene expression from DNA sequence ultimately achieving modest R^2 values across a set of model organisms from bacteria to humans using a deep neural network. While the work is potentially of interest, the overall message and ultimate utility is lost in the delivery of the paper that was very dense to read and understand. Specifically:

The use of R^2 as a metric is not the best to define a model

Response 1.1a

We thank the reviewer for the insightful and constructive comments. Indeed, as the reviewer states, the coefficient of determination R^2 is not the metric to use for defining a model and we are not doing so. To clarify, when training our models we divided the data into training, validation and test sets. Each model was trained on the training set using minimization of squared error criteria (mean squared error, MSE), which is a standard procedure for training regression models¹⁻⁵. Hyperparameter tuning was performed on the validation set and a model with minimal MSE on the validation set was chosen, thus the R^2 was never used to choose or define the model. Then, for final testing evaluation, we again accessed the model with the same MSE criteria to ensure that it did not overfit the data. We acknowledge that indeed it was not clear from the text and we clarified this with additional paragraphs expanding the Methods M1 (page 25, lines 725-756) and M3 sections (page 26, lines 776-783).

In the present study, our models describe and map the relations of input variables to mRNA levels - which are a continuous target variable. Thus, to interpret model performance we measure R^2 on the test data, which is an intuitive metric signifying the percentage of variance that can be predicted from the target variable (mRNA levels) using the information from DNA. This is standard practice for reporting the predictive performance of regression models^{6,7}. The coefficient of determination R^2 has been time-tested and is generally agreed upon by the research and engineering community, especially in the biological sciences, as an intuitively interpretable metric to assess models. This is supported by its use in all of the major publications in the wider field of gene expression research (ie. mapping to a continuous target variable) that also our study contributes to, where R^2 is solely used to assess the performance of shallow (Sharon et al. 2012; Dvir et al. 2013; Cheng et al. 2017; de Boer et al. 2020) as well as deep models

(Agarwal and Shendure 2018; Cuperus, Groves, and Kuchina 2017). The coefficient of determination is defined as $R^2 = 1 - SS_{Residual}/SS_{Total}$ [Eq. 1], where $SS_{Residual}$ is the sum of residual squares of predictions and SS_{Total} is the total sum of squares (specified in manuscript Methods M4). Thus, R^2 describes the proportion of the variance in the dependent variable (target mRNA levels) that is predictable from the independent variables (DNA sequence properties, as defined in Methods M1, Fig S1-1c). It gives the key information on model performance that we are interested in, namely, to what extent are the target mRNA levels encoded in the DNA sequence.

The approach in the present study is in stark contrast to modelling using a discrete variable (termed classification), and for both approaches different methods for assessing the modeling performance are used. We have also considerably rewritten the manuscript to clarify the main message of the work (please see response 1.6) .

Comment 1.1b

and it would be suggested to use alternative statistics approaches to show a goodness of fit such as an F-score as an example.

Response 1.1b

Perhaps the reviewer meant the F-score as a measure of goodness of fit. To clarify, the use of alternative statistics approaches to show a goodness of fit, such as an F-score, is related not to regression but to classification modeling problems, which were not used here. In the machine learning field, the compound metric *F-score* is the harmonic mean of *Precision* (*positive predictive value*, $PPV = TP / (TP+FP)$, where *TP* is the number of true positive predictions and *FP* is the number of false positive predictions) and *Recall* (*true positive rate*, $TPR = TP / (TP+FN)$, where *TP* is the number of true positive predictions and *FN* is the number of false negative). These metrics are not suitable for assessing regression models with a continuous target variable, as one cannot directly measure the true and false positive and negative predictions.

However, perhaps the reviewer meant the F-test as a measure of goodness of fit. The F-test is widely used for statistical inference, e.g. to test the hypothesis whether the model of question fits data significantly better than null model based only on the intercept, i.e. taking the average over the data to make predictions. This is usually a standard practice for inference problems to test for a model's non-zero slope, though it is not widely used in predictive modeling since it is not informative about a model's predictive performance. To make our manuscript accessible to both statistical and machine learning audiences, we have also included *p-values* from an F-test, where relevant, in the main text. Additionally, we have also expanded our assessment of performance measures to include the *mean squared error* $MSE = \frac{1}{n} \sum_{i=1}^n (Y_i - \hat{Y}_i)^2$, which gives the mean of squared differences between actual (measured) Y and predicted \hat{Y}_i values (Table S1-3 and Table S2-1). The MSE incorporates both the variance of the estimators across the different models, i.e. how widely spread the estimates are from one dataset to another, as well as the overall bias - how far off the average estimated value is from the truth. By providing these details, we also want to encourage the genomics and systems biology community to be open about the robustness of their predictive models which are unfortunately often missed in biological studies.

Comment 1.1c

In some cases, the goodness of fit does not seem very strong despite high R^2 (take for example Figure 6B).

Response 1.1c

Please also see the response in 1.1a above. Since we are not assessing the performance of models but relaying the correlation between the measured GFP fluorescence and predicted expression levels, we show Pearson's correlation coefficient r , which is also the standard metric to assess and report correlation between variables in the field ^{8,9}. In general, since R^2 can be derived as an approximation of the correlation coefficient r (then termed r^2), also r is approximately the square root of R^2 (in Fig 6B this would correspond to ~ 0.42). However, one metric (R^2) is defined and used for assessing the performance of a model based on variance analysis of the true versus predicted variables, and the other (r) is used to assess the strength of a (linear) relation between two independent variables. Therefore, we are trying to clearly differentiate between reporting model performance with R^2 and reporting correlations between variables with r .

Comment 1.2

A comparison of this machine learning approach to other approaches (including things like the RBS calculator and other predictive approaches) should be used to determine the appropriateness of this learning method and the resulting models.

Response 1.2

As the reviewer points out, we have systematically investigated the appropriateness of the proposed method. The main model organism that is used to build and interpret models in the study is *Saccharomyces cerevisiae*. For yeast, after a comprehensive literature search, we found that no complementary predictive approach exists that could be directly compared, especially based on DNA sequence and predictive of mRNA expression levels. This is also the case, since no study has until now attempted to merge such a large amount of mRNA sequencing experiments - thus uncovering the relatively stable conservation of mRNA levels across genes and conditions - and attempted to relate them to gene coding and regulatory sequences. However, scientific publications have reported different R^2 values of models based on certain specific coding or non-coding regulatory regions, such as codon frequencies^{10,11}, promoters^{12,13} and UTRs^{14,15}, the highest of which are specified in Fig S1-1B. These publications also served as the basis for our selection of the most informative properties to use in modeling (see Fig S1-1C).

Our aim was though not a direct comparison with published performance metrics, since the scores obtained in our study are considerably higher than in any known published studies. This can be attributed to the generalization of our approach that uses both (i) all available RNA-Seq data and not just a single experiment, and (ii) improved modeling input variables, by taking the whole gene regulatory structure into account, something that has not been done until now. Thus, to justify appropriateness of our learning method and resulting models, we systematically compared the predictive performance for mRNA expression, resulting from every regulatory region studied so far (i.e. promoter, terminator, UTRs, CDS and their combinations) using both widely used classical machine learning approaches as well as state-of-the-art deep neural networks (Fig 2A, B), which demonstrated the validity and the power of the present approach. Analogously, we analysed 6 model organisms (*Escherichia coli*, *Arabidopsis thaliana*, *Drosophila melanogaster*, *Danio rerio*, *Mus musculus* and *Homo sapiens*) to verify and confirm that the observed phenomenon (the high predictive power of regulatory and coding DNA sequence for mRNA expression levels) was indeed detectable across the whole tree of life (Fig 1H).

Furthermore, since it would be unreasonable to establish the appropriateness of our learning method and the resulting models based only on comparison of model performance metrics, and to further justify the biological validity of the results, we (i) performed a comprehensive analysis of DNA sequence motif occurrences and co-occurrences across the regions to find the underlying principles that enable our models to perform so well (Fig 4F, Results chapter 4), and (ii) experimentally verified the predictive power of the approach and its usefulness for guiding experiment design (Fig 6B, Results chapter 6).

Also as the reviewer suggested using *E.coli* (RBS calculator implementation is limited to *bacteria*) we have tested and compared results with the specified RBS calculator^{16,17} algorithm, which predicts mRNA translation levels based on input mRNA sequence (RBS calculator online prediction tool was used: <https://www.denovodna.com/software/>). However, our method and the RBS calculator aim to predict fundamentally different phenomena. Since the RBS calculator uses a different target variable (mRNA translation rate) than the one used in our study (mRNA expression level), we could not directly compare the predictions. Instead, we analysed the correlations between RBS calculator predictions to the data used in our study - average of experimentally observed mRNA levels from thousands of gene expression studies (Figure R1). Indeed, since here we are working with mRNA transcript levels as opposed to the mRNA translation rates with the RBS calculator, and due to the highly different nature of the data and modeling (thousands of experiments and deep learning vs. thermodynamic models) neither the experimentally derived mRNA values used for modeling (Fig R1A), nor our deep learning-predicted mRNA values (Fig R1B) correlate well (Pearson's $r = 0.128$, $p\text{-value} = 9.4e-2$) with the RBS calculator predictions. The deep learning model however does well what it was trained for, namely, to predict median mRNA levels based on gene regulatory structure sequence and properties (Fig R1C, Methods M1 and M2). Since the aim of our study was not directly related to protein translation rates, but instead mRNA levels, and in order to not confuse the reader, since we already describe complex modeling efforts, we deem that these analyses should not be included in the present manuscript.

Figure R1-1. Correlation analysis of results with RBS calculator and the *Escherichia coli* model organism. Red lines denote least squares fits. (A) Comparison of median mRNA values used for modeling in our study and RBS calculator-predicted translation rates. (B) Comparison of predicted mRNA values with our deep learning model and RBS calculator-predicted translation rates. (C) Actual vs deep learning-predicted mRNA levels.

Comment 1.3

The predictive power of the approach to a de novo sequence would be suggested.

Response 1.3

Please see the Response 1.1a. In order to test the model's predictive power on new sequences that it had not seen before, we used a held out test set (10% of protein coding genes, >400 data points), which is a standard approach for testing the predictive performance of machine learning models^{3,6,7}. We have updated the manuscript text to make this more clear (page 5, lines 139-156, Fig 1).

We also showed that the model mRNA level predictions are in high agreement with GFP measurements in synthetic constructs (Figure 1G and S1-5B) with two independent datasets comprising thousands of data points, which were not used for training the model (data not seen by the model). Furthermore, we have now additionally reviewed the literature for verified synthetic yeast regulatory sequences that could be used for additional “*de novo*” testing. However, all of these sequences are 250 bp or shorter (e.g. synthetic promoters and corresponding scaffolds) and span merely one of the four regulatory regions^{13,15,18–26}, whereas our models were trained on larger sequences, spanning 1000 bp of promoters and altogether 2150 bp (even the minimal model developed for experimental purposes in the final chapter of results was based on 1000 bp of sequence). Nevertheless, we were able to prepare *de novo* datasets by randomly shuffling the sequences and preserving their dinucleotide content, in order to verify that DNA sequences contain specific expression grammar that is not merely a function of the GC content. As expected, we observed that the model predictive capacity breaks down with the non-informative random sequences that carry no regulatory information (Fig R2A,B: over 2.2-fold decrease in median predicted expression levels compared to experimentally measured or non-random predicted ones).

Figure R1-2. De novo testing of the yeast model with randomly shuffled and cross-organism sequences. (A) Predicted expression levels and (B) coefficient of variation R^2 with randomly shuffled sequence data conserving dinucleotide content, compared to non-randomized sequences and the experimentally measured expression levels.

Comment 1.4a

The authors set boundaries for the various promoter, terminator etc elements that (in some organisms) would overlap with neighboring genes. How often was this the case?

Response 1.4a

Since *Saccharomyces cerevisiae* was the central model organism of our study, while the other model organisms used to support the main finding, the boundaries of the regulatory regions were selected based on an overview of current published yeast studies (Figs 1D and S1-1C: 1000 bp, 300 bp, 350 bp and 500 bp for promoters, 5'UTRs, 3'UTRs and terminators, respectively).

As suggested by the reviewer, we have analysed the overlap between the promoter and terminator regions of genes sorted according to the order of CDS occurrence in the yeast genome (across the 16 chromosomes). By testing which intervals of the regulatory regions overlap, we observed that 55% of genes have an overlap between promoter and terminator regions with their neighbors (Figs R3A, B: ratio of genes with overlaps out of all genes was 0.55). Besides 44% of genes that overlap with the nearest neighbor gene (ie. distance is one gene away), due to laying on opposing DNA strands, 11% of genes overlap with their second or third nearest neighbor (ie. distance is 2 to 3 genes away, Fig R3B: mean gene overlap distance is 1.21, median is 1). Next, we analysed also the other model organisms using the same boundaries (Fig 1D) and metrics (ratio of overlapping genes, ratio of genes overlapping with 1 gene away or more than 1 gene away, avg. distance, Fig R3B). As expected according to current knowledge²⁷⁻²⁹, with increasing organism complexity, thus increasing genome size and decreasing genomic complexity (as measured by the number of genes per Mbp), the ratio of overlapping genes decreased (Fig R3B), which was also the case with the other metrics. Indeed, a very high correlation was observed between the ratio of overlapping genes and genomic complexity (Pearson's $r = 0.989$, p -value $< 2.3e-5$, Fig 3C). The number of overlapping genes rose to 80.7% for *E. coli* and fell to 2.2% with *H. sapiens*, with the mean distance of overlapping genes rising to 3.90 and falling to 1.01, respectively. Organisms of lower genome complexity, such as bacteria and yeast, have more compact genomes with less non-protein coding regions, and thus more overlap between regulatory elements^{28,30} with less space for the more complex and distant regulation (e.g. enhancers that regulate gene expression from thousands of bps away) found in more complex organisms from plants to human³¹⁻³⁴. However, despite these overlaps in the regulatory regions and consequently a sharing of regulatory signals between some of the genes, expression levels between sets of overlapping genes can be highly variable, as with yeast we measured a median standard deviation of 40.3 TPM that reached a maximum of 12479 TPM (Fig 4D). This supports the appropriateness of our modelling approach and the selected gene sequence bounds.

We thank the reviewer for this suggestion and we have included these new results in the manuscript as a supplementary figure (Fig S1-9).

Figure R1-3. Analysis of gene overlap. (A) Dot-plot of overlapping genes on chromosome 16 in yeast showing approximately half of genes promoters and terminators overlap with, on average, their first neighbor gene (80%), and up to 3 neighboring genes (20%). (B) Gene overlap measures across the model organisms including the ratio of genes with overlaps out of all genes (ratio_overlap), mean distance between overlapping genes (mean_distance), ratio of genes overlapping with their nearest neighbor gene (ratio_dist=1) and ratio of genes overlapping with genes farther than their first nearest neighbor (ratio_dist>1). (C) Correlation analysis between ratio of overlapping genes and genomic complexity. (D) Variation of gene expression (median TPM) observed across the overlapping sets of genes in yeast.

Comment 1.4b

How much influence does this exact window have on the determination of function from sequence?

Response 1.4b

This is indeed an interesting question and was explored in our analysis, where we used sizes of regulatory regions giving optimal model performance. This was however not initially reported, as the region size was not the most crucial factor determining model performance (Fig R1-4A). When testing the effect of the region size on model predictive performance, we found that a 50% decrease in the size of the regions led to a 7% decrease in model performance, with the consecutive decreases having a larger effect due to the depleting amount of information left in the regions. Thus, despite the overlaps found in certain regulatory regions (Fig R1-3, see above response 1.4a), regulatory signals important for expression of their specific gene are still interspersed across the whole regions (Fig 3A: relevance z-scores of all parts of the regulatory sequences surpassed 2 standard deviations). All regions carry information important for accurate predictive modelling (Fig 2B: model performance consistently increases with additional regions), especially in the form of co-occurring sets of DNA regulatory motifs uncovered across all the regions (Fig 4F).

Our selection of the most informative properties and sequence regions to use in modeling (see Fig 1D, Fig S1-1C) was based on a comprehensive overview of the published literature including experimental and modeling studies. This included the studies reporting the different R^2 values of models based on certain specific coding or non-coding regulatory regions, such as codon frequencies^{10,11}, promoters^{12,13} and UTRs^{14,15}, as mentioned in response 1-2 above (see Fig S1-1c for additional references). The rational selection of region sizes, sufficiently large to cover the most important regulatory signals (Fig 1D: spanning altogether 2150 bp of regulatory sequence per gene), was also here possible due to using deep learning methods. Apart from enabling the use of DNA sequence as input, due to their capability to learn optimal data representations themselves (and thus interpret DNA regulatory motifs)³⁵ the performance of deep methods does not suffer with increased amounts of input data as with classical shallow machine learning.

An additional confirmation that the selected region sizes were sufficient was obtained with another approach. The knowledge obtained with further analysis of the models and input data, namely, uncovering that all regions interact and measuring the relevance of each position in the regulatory regions, enabled us to test a more rational approach of constraining the regulatory regions by selecting only the parts of the regions with the most pronounced relevance scores (Fig 3A: selection of 400 bp, 100 bp, 250 bp and 250 bp of promoters, 5'UTRs, 3'UTRs and terminators, respectively). This indeed led to models that achieved almost the same performance as the original ones (Fig R4B) based on approximately half shorter regulatory regions (altogether 1000 bp compared to 2150 bp of the full models). The improvement of this model over the one with exactly 50% shorter regions described above (Fig R1-4A) however, was likely due to the remaining longer part of the 3'UTR region (250 bp of 350 bp), which had a joint effect with the other regions leading to the slightly improved performance.

We have included these results in the manuscript as a supplementary figure (Fig S1-10) and added appropriate descriptions in the main text (page 8, lines 216-222 and page 26, lines 776-783).

Figure R1-4. Analysis of the effect of regulatory region window size on the determination of function from sequence. (A) Effect of decreasing the sizes of the input regulatory sequences on model performance. (B) Rational design of a smaller 1000 bp model according to the analysis of position-specific relevance of the input sequence in the full 2150 bp model (see Fig 3A in the manuscript), where only the combination of the most relevant region sizes were used, namely 400 bp, 100 bp, 250 bp and 250 bp of promoters, 5'UTRs, 3'UTRs and terminators, respectively.

Comment 1.4c

How much would a promoter (as an example) taken out of context be influenced by the genomic context? Can this be predicted by the model?

Response 1.4c

This is an interesting question and it was indeed addressed in the paper. Please refer to Results chapter 2 (Fig 2) in the main text (pages 8-9, lines 208-273), where we compared the context of regulatory regions on model predictive performance. To assess the effect of the genomic context on regions taken out of context, meaning that we swap a certain promoter or terminator region with another variant from a different gene, we performed a large systematic analysis in Results chapter 5 in the main text (pages 17-21, lines 455-610). Here, both predictions (Fig 5) and experimental validations (Fig 6) demonstrated that the model is indeed sensitive to region perturbations, and that the genomic context has a large and differing effect on regions taken from alternate contexts (ie. different genes). In addition, we have strived to improve the main text in Results chapter 5 (pages 17-21, lines 455-610) and accompanying Figures 5 and 6 to further clarify the points raised by the reviewer.

Comment 1.5

Extracted motifs such as those in Figure 3 should be head-to-head compared to other work in the literature that have extracted similar phenomena.

Response 1.5

We agree with the reviewer and we have tried applying other methods that have extracted similar phenomena to extract DNA regulatory motifs with our models, even before using the present approach. Indeed, we presumed that existing methods, such as DeepLift³⁶, TF-Modisco³⁷ and DeepExplain³⁸ would enable comparisons between motifs extracted from different models. However, despite our efforts to apply these methods to our models as well through the help of the method's authors (A. Shrikumar from Kundaje lab, Stanford), we were unable to obtain motifs due to the underlying differences between our models and the models that those methods are developed for. Specifically, the existing methods for extracting sequence motifs from convolutional neural networks (CNN) are only developed for classification methods with discrete target variables, as far as we are aware. This includes, for instance, transcription factor binding models with general 'binding' or 'no-binding' events (or some discrete states within)^{37,39}. On the other hand, our regression models predict a continuous variable based on which the existing methods cannot extract appropriate information to reconstruct motifs. Consequently, according to our understanding and survey of the field, we choose a well established method that predicts the relevance of an input for the target variable based on input occlusions^{38,40,41}. The method had already been successfully used in genomics in the classification setting⁴¹, and we implemented it here to enable extracting motifs from the regression models trained in our setting. To our knowledge, this is the first implementation that indirectly infers motifs and is related to a phenotypic outcome (gene expression), whilst it is not dependent on direct modeling of binding events that would be biased to the provided data (such as e.g. ChIP-Seq data), which would frequently result in alike motives related to TF binding regions.

We therefore compared our motifs to the actual TF-binding motifs from the Jaspar and Yeastract databases that represent standard motif references and found hundreds of significant hits to actual known TF binding sites (BH adj. p -value < 0.05). Additionally, we verified that our motifs comprised also additional known DNA motif grammar (Fig S1-1), which comprises, besides the classic TF binding sites, also (i) additional regions and binding sites that include multiple well-known sequences, such as TATA boxes⁸, Kozak sequences²⁶, A and T-rich sites, positioning and efficiency elements²⁰, spread out across the different regulatory regions, respectively (Fig 3F), (ii) elements related to nucleosome positioning^{22,23} (Fig S3-3), as well as (iii) regions adjacent to the binding sites that create a larger protein recognition and binding template and potentially contained conserved DNA physicochemical properties⁴²⁻⁴⁴, though are predictable from sequence^{43,45} and thus learnable by the deep models.

Comment 1.6

The paper needs to be completely revised for clarity of reading.

Response 1.6

We have strived to improve the overall clarity and readability of the manuscript. Changes in the text are highlighted as per the editors instructions.

Points that were addressed include:

1. Improvement of overall readability of all sections:
 - a. avoiding long and intricate phrases (with many appositions),
 - b. preference to be overly explicit,
 - c. less dense sections of the results.
2. Clarification of the objectives of the paper:
 - a. which organisms and data used and where,
 - b. clear aims of each analysis,
 - c. overall conclusions and applicability (ie. not a methods or 'utility' paper but theoretical study showing interesting biological findings).
3. Reference to methods section was expanded to methods subchapters and methods were revised as per the reviewers instructions (pages 25-29, lines 726-870).
4. Further testing and explanation of chosen region boundaries:
 - a. amount of effect of window have on the determination of function from sequence (Fig S1-10),
 - b. overlap of regions (Fig S1-9).
5. Clarification of the effect of the contribution of each DNA region to the model predictive power (page 8, lines 216-222 and page 26, lines 776-783,, Fig S1-10).
6. Revision of results and discussion related to the regulatory and coding region coevolution findings and their indications (pages 8-9, lines 239-273 and page 23, lines 654-665, Fig S2-3).
7. Discussion on the models being more capable of predicting downregulation compared to upregulation (page 24, lines 704-717, Fig S6-6).
8. Revision of the experimental results section (pages 20-21, lines 578-615, Fig 6C,D).
9. Multiple additional corrections and improvements (see all Responses).

Referee 2

Comment 2.1

Reviewer #2 (Remarks to the Author):

The authors propose a very interesting computational method to try and deduce the DNA regulatory code underlying gene expression changes in several organisms. They are able to explain a high percentage of the gene expression changes using a deep learning approach and look into the effect of (combinations of) different parts of the regulatory elements on gene expression.

While I think this is a very interesting paper with important findings of interest to a wide audience, I do think that the paper needs more attention, in particular with regard to overall readability and also addressing/mentioning the drawbacks.

Response 2.1

We are glad the reviewer finds our study interesting and important and we thank the reviewer for insightful and constructive comments. We have strived to improve the overall clarity and readability of the manuscript. Changes in the text are highlighted as per the editors instructions. Points that were addressed include:

1. Improvement of overall readability of all sections:
 - a. avoiding long and intricate phrases (with many appositions),
 - b. preference to be overly explicit,
 - c. less dense sections of the results.
2. Clarification of the objectives of the paper:
 - a. which organisms and data used and where,
 - b. clear aims of each analysis,
 - c. overall conclusions and applicability (ie. not a methods or 'utility' paper but theoretical study showing interesting biological findings).
3. Reference to methods section was expanded to methods subchapters and methods were revised as per the reviewers instructions (pages 25-29, lines 726-870).
4. Further testing and explanation of chosen region boundaries:
 - a. amount of effect of window have on the determination of function from sequence (Fig S1-10),
 - b. overlap of regions (Fig S1-9).
5. Clarification of the effect of the contribution of each DNA region to the model predictive power (page 8, lines 216-222 and page 26, lines 776-783,, Fig S1-10).
6. Revision of results and discussion related to the regulatory and coding region coevolution findings and their indications (pages 8-9, lines 239-273 and page 23, lines 654-665, Fig S2-3).

7. Discussion on the models being more capable of predicting downregulation compared to upregulation (page 24, lines 704-717, Fig S6-6).
8. Revision of the experimental results section (pages 20-21, lines 578-615, Fig 6C,D).
9. Multiple additional corrections and improvements (see all Responses).

Comment 2.2

Major points:

- I found the first part of the results (pg 6, 7 model definition) a bit hard to get the exact details of how the model was obtained. At various times, the authors refer to the methods, but was unable to unambiguously find the section in the methods section. It would be really helpful if the authors provide a more extensive methods (or maybe supplemental methods) section with regard to the data processing (maybe split it up by data type?) and parameters used for fitting. This was now an exercise in trying to match the main text with the methods and supplemental figures, without a clear guidance.

Response 2.2

We thank the reviewer for raising this point. We have updated the methods section naming each section with corresponding M1, 2, ..., 10 to clearly point to each section from within results. Section Methods M1 (page 25, lines 725-756) was improved by separating the text according to different data types and clarification of the data processing descriptions.

Comment 2.3

- Throughout the text it remains a bit unclear when the authors switch from the analyses performed for all seven organisms to the indepth analyses performed using yeast data. It would help alot if the authors make much clearer that the majority of the results and conclusions are based on yeast. Only the global variance estimates where verified in other organisms, which also clearly showed that more complex organisms are harder to predict.

Response 2.3

We see that this was not clearly explained in the initial version of the manuscript. The different model organisms are summarized in the abstract and introduction sections. The main model organism that is used to build and interpret models in the study is *Saccharomyces cerevisiae*, whereas the additional 6 model organisms (*Escherichia coli*, *Arabidopsis thaliana*, *Drosophila melanogaster*, *Danio rerio*, *Mus musculus* and *Homo sapiens*) were used to verify that the observed phenomenon (the high predictive power of regulatory and coding DNA sequence for expression levels) was indeed detectable across the whole tree of life. We have clarified the text in the introduction, results and discussion sections, to specify that the analysis is performed with yeast and specifically define where the additional model organisms are used, which was only in the final paragraph of first section of results (pages 6-7, lines 177-192).

Comment 2.4

- I am not convinced by the co-evolution analyses. While I'm not an evolutionary biologist, claiming that regulatory regions co-evolve with the corresponding coding regions based on just one correlation analysis seems a bit overenthusiastic. How does for instance the correlation look like with other parts of the genome that are not directly related, is that a much lower number?

Response 2.4

In the coevolutionary analysis, we measure the correlation between the mutation rates in regulatory regions (promoters and terminators) and the corresponding coding regions. Indeed, not including any negative controls in the analysis was an oversight on our part and we think this is a very good suggestion. We tested both the effects of method sensitivity as well as the potential for the coevolution across larger parts of the genome. Therefore, considering that yeast has a very compact genome with over half of the gene's regulatory regions overlapping (Fig S1-9), a rational test to use as control was to analyse if there is any correlation between the regulatory and coding regions of non-related genes (achieved by random shuffling of the regulatory regions). The negative control shows that no correlation is observable across the non-related regions (Pearson's $r \sim 0$, p -value > 0.05).

Additionally, we expanded our overview of published studies and findings. In higher eukaryotes, multiple lines of evidence show contributions of positive selection in *cis*-regulatory regions including promoters⁴⁶, transcription factor binding sites^{47,48} and enhancers^{49,50}, besides purifying selection in their maintenance⁴⁷. In fact, approximately half of all functional variation was found in non-coding regions⁵¹. Orthologous genes thus display a coupling of protein and regulatory evolution⁵², suggesting that selective pressure on gene expression and protein evolution is quite similar and persists for a significant amount of time following speciation⁵² and that even regulatory sequence can evolve to fine-tune expression levels⁵³. In yeast, due to the hypervariable non-coding sites that could result from selection on regulatory mutations⁵⁴, a similar coupling of gene mutation rates with the gene's expression levels was found⁵⁵ as well as evidence to causally link differences in gene expression to variation at individual regulatory nucleotide positions⁵⁶. Indeed, expression levels deviating from the parental range in a yeast strain were found to occur through novel regulatory-gene mutational interactions⁵⁷.

Considering these publications and based on the strength and significance of the measured correlation in our study (Fig 2E, F), we assume that our results give evidence for the coevolution hypothesis. However, we fully agree that the way these results were presented previously without controls was not appropriate, and we have now included the new results as a supplementary figure S2-3, made a more appropriate presentation of the findings and their indications in the manuscript (pages 8-9, lines 239-273) as well as expanded the discussion section to include the presently published findings and their references (page 23, lines 654-665).

Figure R2-1. Analysis of coevolution of regulatory and coding regions in orthologous genes of 14 yeast species. Red lines denote least squares fits. (A) Control analysis of evolutionary substitution rates in promoter vs. coding regions, where the regions were randomly mismatched. (B) Control analysis of evolutionary substitution rates in terminators vs. coding regions, where the regions were randomly mismatched. (C) Evolutionary substitution rates in terminators vs. promoter regions. (D) Control analysis of evolutionary substitution rates in terminators vs. promoter regions, where the regions were randomly mismatched.

Comment 2.5

- The authors seem to ignore the fact that their model is much more capable of predicting downregulation compared to upregulation (pgs 7 and 18). This warrants some thoughts in the discussion about the implication of their model predictions for future experiments and a potential explanation why this might happen (can imagine that down more often equates to off and therefore is easier to predict than up).

Response 2.5

As the reviewer pointed out, with the experimental testing (Fig 6B) the model was on average 1.6-fold more capable of predicting downregulation than upregulation (based on comparing fluorescence intensity medians). A note, we have recomputed and replotted Figures 6C and D, due to a computation error in the previous article version, though the overall results and conclusions remain similar, and on average a 24% decrease in gene expression is observed with weak promoters compared to a 15% increase with strong ones (Fig 6C, D). An explanation for the higher predictive power for downregulation compared to upregulation could indeed be related to the selection pressure of highly expressed genes. For multiple organisms, including viruses, it has been shown that very highly expressed genes have lower sequence divergence^{58,59} and thus altering highly optimized sequences would likely result in downregulation. Along these lines we hypothesize the existence of a regulatory grammar 'fitness landscape', similar as with other molecular, e.g. protein, fitness landscapes⁶⁰. Grammar optimized for increased expression represents peaks in the landscape (is potentially more 'focused' and rare), whereas some basal lower level expression is represented by the valleys with many more variations of the grammar. The exception is possibly with very low expression, which again is more defined, so represented by an inverted valley-to-peak landscape. Accordingly, when altering the regulatory information in the sequences, by shuffling the input sequences preserving their dinucleotide content, we observe that average model predictions on shuffled sequences are over 2-fold lower than with original non-shuffled ones (Fig R2-2A, B). This suggests that a certain basal level of expression exists, lower than the organism average but still above zero (Fig R2-2A: predicted at ~2-fold lower than the median expression level, 64.5 TPM). This makes sense also from an evolutionary perspective, as for both very low and high expression the regulatory grammar must specially evolve to define these levels (Fig R2-2C). On the other hand, it can be expected that for the basal expression level regulation is less specific, possibly either 'turned off' or comprising a more diverse grammar. This gives some explanation why it might be easier for the model to predict downregulation than upregulation. The implications of these findings for future experiments are that (i) possibly separate models for different classes of expression as well as accounting for different conditions might give better results for specific predictions, as well as (ii) more computational and experimental work is required to decipher the evolutionary strategies of regulatory grammars and define the properties of underlying basal and targeted-evolved regulation. We have thus included these results in the manuscript (Supplementary figure S6-6) and expanded the discussion section as suggested (page 24, lines 704-717).

Figure R2-2. Prediction of a basal level of *S. cerevisiae* gene expression and a hypothesized regulatory grammar fitness landscape. (A) Predicted expression levels with randomly shuffled sequences with conserved dinucleotide content, compared to non-randomized sequences and the experimentally measured expression levels. Median predicted expression level with shuffled sequences was over 2-fold lower than with original ones, at 64.5 TPM. (B) Hypothesis on the potential evolutionary fitness landscape of regulatory grammar that can be inferred from our results.

Comment 2.6

- I am intrigued by the difference they found between the shallow and deep modeling with regard to the contribution of the regulatory regions. While the shallow modeling is unable to explain a large part of the variance, the deep modeling is. When combined with the codon frequencies, the difference between the two approaches is much smaller. The authors also indicate that they can predict the codon usage using the regulatory code. How then can such a big difference between the two modeling approaches exist? The shallow modeling suggests that the codon frequency is much more important compared to the regulatory regions, while the deep modeling is assigning a much higher contribution to the regulatory regions.

Response 2.6

Shallow modeling approaches have the limitation that they cannot decode the information in a DNA sequence directly and thus rely on human feature engineering^{35,61}. Since they cannot learn sequence relations such as motif occurrence and co-occurrence, they require the use of some quantifiable feature, such as k-mer frequencies, as a representation of the sequence properties. K-mers, despite representing the different motifs inside the sequence and thus giving a representation of the motif landscape to the model^{62–65}, are not an optimal representation of the complexity of information encoded in the DNA, such as the regulatory grammar uncovered in our study. In contrast, deep learning models can themselves interpret the features directly from data^{35,66}, where specifically convolutional neural networks (CNN) can learn to recognize sequence motifs as well as their co-occurrence across the DNA sequence^{15,41,67,68}. Deep neural networks can thus be viewed as an expansion of shallow models, being capable of learning practically everything that the classical ML models can, but also learning the best underlying representations of the features. Therefore, although both procedures gave good and very similar results with the codon frequencies, since both can directly use these features to build models, with deep modeling also the information encoded in the regulatory regions could be deciphered and was thus used for the predictions (manuscript Fig 2A). Since, due to the potential coevolution (Fig 2E, F and R2-1), there is a large overlap of the information in the coding and regulatory regions, the increase in deep model performance when using both coding and regulatory regions was not as drastic (Fig 2A: R^2 increased from 0.690 with coding to 0.816 with coding+regulatory) as when comparing shallow and deep models using only regulatory region sequence features (Fig 2A: R^2 increased from almost 0 to ~0.5). Therefore, the big difference between the two modeling approaches occurs in case of comparing regulatory regions, where with shallow models, k-mers were used as features to represent the DNA sequence, and with deep models, the DNA sequence could be used directly. Indeed, the shallow modeling suggests that the codon frequency is much more important compared to the regulatory regions, which is in case of the shallow models. In contrast, the deep modeling is assigning a high contribution to the regulatory regions, but since the information in these regions overlaps with the information in the coding regions, the increase in predictive power of the

models with regulatory + coding regions compared to models with only coding regions is not as pronounced as would be expected solely on the amount of information encoded in the regulatory regions. As the reviewer pointed out, this was not clear in the previous version of the manuscript, and we have thus updated the manuscript text to better communicate this finding (page 8, lines 209-230).

Comment 2.7

minor remarks:

- pg 6, FigS1-2, in contrast to what they authors indicate, I only see metabolic processes, transport and stress response in the enrichment analysis.

Response 2.7

Thank you for pointing out this mistake, which was likely lingering from a previous version of the results. We have fixed this in the text (page 5, lines 134-137) as you have suggested.

Comment 2.8

- Fig1G, Why use a pearson r here, while in all other subpanels an R^2 is used. This is confusing.

Response 2.8

The difference was that plot (F) showed a standard assessment of modeling results using R^2 and true (target variable) vs predicted values of a held out test dataset, whereas the other plot (G) shows a non-modelling assessment of how model predictions correlate to published GFP experimental values, using experimental vs. predicted values on the subset of available data, where we were not training models based on these data.

To clarify further, our models describe and map the relations of input variables to mRNA levels - which is a continuous variable. Thus, for the interpretation we present model performance as R^2 on the test data, which is an intuitive metric signifying the percentage of variance that can be predicted from target variable (mRNA levels) using the information from DNA. The coefficient of determination is defined as $R^2 = 1 - SS_{Residual}/SS_{Total}$ [Eq. 1], where $SS_{Residual}$ is the sum of residual squares of predictions and SS_{Total} is the total sum of squares (specified in manuscript Methods M4). Thus, R^2 describes the proportion of the variance in the dependent variable (target mRNA levels) that is predictable from the independent variables (DNA sequence properties, as defined in Methods M1, Fig S1-1C), which gives the key information on model performance that we are interested in, namely, to what extent are the target mRNA levels encoded in the DNA sequence.

For Figure 1G, since we are not assessing the performance of models but relating the correlation between the measured GFP fluorescence and predicted expression levels, we show Pearson's correlation coefficient r , which is also the standard metric to assess and report correlation between variables in the field^{8,9,69}. In general, since R^2 can be derived as an approximation of the correlation coefficient r , also r is approximately the square root of R^2 (in Fig 6B this would correspond to ~ 0.42). However, one metric (R^2) is defined and used for assessing the performance of a model based on variance analysis of the true versus predicted variable, and the other (r) is used to assess the strength of a (linear) relation between two independent variables. Therefore, we were trying to clearly differentiate between reporting model performance with R^2 and correlations between variables with r .

Comment 2.9

- pg 9, the authors claim that the combination is additive "Each DNA region thus additively contributed ..." This warrant some further explanation. Effects can also be non-additive (multiplicative for instance), why/how did the authors conclude this?

Response 2.9

Yes, after reassessing these statements, we find that this wording was not appropriate according to the findings, since we did not assess the exact nature but observed the overall combined result. To avoid the ambiguity of the meaning of the word "additive", which as the reviewer points out in a mathematical sense would require justification of the nature of the interaction, we thus rephrased the term to "jointly", as a combined effect of regions contributing to the predictions. The meaning that we think is appropriate is that the regions 'jointly' contributed to the increase in predictive performance, which implies some 'combination' and 'addition', but is generic enough not to be misleading.

Comment 2.10

- pg 22, discussion. There, the authors indicate that the biological variation due to different conditions is negligible compared to the gene expression difference encoded in the genome. I think the authors have to be a bit more careful here with their statement. I would disagree that biological variation is negligible. In essence, the parameters they choose to include in their model (motifs, regulatory sequence) are effectively ways for the system to encode responses to different environmental changes through transcription factor binding, chromatin rearrangements, etc and therefore should be considered an indirect way of environmental changes on gene expression levels.

Response 2.10

Yes, the term biological variation indeed often encompasses more than is (strictly) assessed here and also often implies dynamics. We of course did not mean to neglect biological system dynamics and, as the reviewer points out, the result of observed expression levels as a response to different environmental conditions. However, the term used in the discussion strictly relates to the gene expression variation across thousands of experiments and conditions analyzed in the present study (Fig 1A), where for each gene the variability across the conditions was much smaller than the entire dynamic range of expression levels across all genes. Since the median value across the conditions (per gene) is the target variable, this defines what our models can 'learn' to predict, where any condition-specific changes per gene were likely not captured by the current models, as they were not included in their training data. We therefore fixed the wording to avoid strong phrases like 'negligible' to improve the clarity of this text (page 22, lines 628-630).

Document references

1. Tian, Q. *et al.* MRCNN: a deep learning model for regression of genome-wide DNA methylation. *BMC Genomics* **20**, 192 (2019).
2. Lathuilière, S., Mesejo, P., Alameda-Pineda, X. & Horaud, R. A Comprehensive Analysis of Deep Regression. *IEEE Trans. Pattern Anal. Mach. Intell.* 1–1 (2019).
3. Géron, A. *Hands-On Machine Learning with Scikit-Learn, Keras, and TensorFlow: Concepts, Tools, and Techniques to Build Intelligent Systems.* ('O'Reilly Media, Inc.', 2019).
4. Kuhn, M. Building Predictive Models in R Using the caret Package. *J. Stat. Softw.* **28**, 1–26 (2008).
5. James, G., Witten, D., Hastie, T. & Tibshirani, R. *An Introduction to Statistical Learning: with Applications in R.* (Springer, New York, NY, 2013).
6. Kuhn, M. & Johnson, K. *Applied Predictive Modeling.* (Springer, New York, NY, 2013).
7. Hastie, T., Tibshirani, R. & Friedman, J. *The Elements of Statistical Learning: Data Mining, Inference, and Prediction, Second Edition.* (Springer Science & Business Media, 2009).
8. Lubliner, S. *et al.* Core promoter sequence in yeast is a major determinant of expression level. *Genome Res.* **25**, 1008–1017 (2015).
9. Wilhelm, M. *et al.* Mass-spectrometry-based draft of the human proteome. *Nature* **509**, 582–587 (2014).
10. Cheng, J., Maier, K. C., Avsec, Ž., Rus, P. & Gagneur, J. Cis-regulatory elements explain most of the mRNA stability variation across genes in yeast. *RNA* **23**, 1648–1659 (2017).
11. Neymotin, B., Ettore, V. & Gresham, D. Multiple Transcript Properties Related to Translation Affect mRNA Degradation Rates in *Saccharomyces cerevisiae*. *G3* **6**, 3475–3483 (2016).
12. Espinar, L., Schikora Tamarit, M. À., Domingo, J. & Carey, L. B. Promoter architecture determines cotranslational regulation of mRNA. *Genome Res.* **28**, 509–518 (2018).
13. Sharon, E. *et al.* Inferring gene regulatory logic from high-throughput measurements of thousands of systematically designed promoters. *Nat. Biotechnol.* **30**, 521–530 (2012).
14. Dvir, S., Velten, L., Sharon, E. & Zeevi, D. Deciphering the rules by which 5'-UTR sequences affect protein expression in yeast. *Proc. Natl. Acad. Sci.* **110**, E2792–E2801 (2013).
15. Cuperus, J. T., Groves, B. & Kuchina, A. Deep learning of the regulatory grammar of yeast 5' untranslated regions from 500,000 random sequences. *Genome Res.* **27**, 1–10 (2017).
16. Salis, H. M., Mirsky, E. A. & Voigt, C. A. Automated design of synthetic ribosome binding sites to control protein expression. *Nat. Biotechnol.* **27**, 946–950 (2009).
17. Espah Borujeni, A. *et al.* Precise quantification of translation inhibition by mRNA structures that overlap with the ribosomal footprint in N-terminal coding sequences. *Nucleic Acids Res.* **45**, 5437–5448 (2017).
18. de Boer, C. G. *et al.* Deciphering eukaryotic gene-regulatory logic with 100 million random promoters. *Nat. Biotechnol.* **38**, 56–65 (2020).
19. Sharon, E. *et al.* Probing the effect of promoters on noise in gene expression using thousands of designed sequences. *Genome Research* vol. 24 1698–1706 (2014).

20. Curran, K. A. *et al.* Short Synthetic Terminators for Improved Heterologous Gene Expression in Yeast. *ACS Synth. Biol.* **4**, 824–832 (2015).
21. Redden, H. & Alper, H. S. The development and characterization of synthetic minimal yeast promoters. *Nat. Commun.* **6**, 7810 (2015).
22. Curran, K. A. *et al.* Design of synthetic yeast promoters via tuning of nucleosome architecture. *Nat. Commun.* **5**, 4002 (2014).
23. Morse, N. J., Gopal, M. R., Wagner, J. M. & Alper, H. S. Yeast Terminator Function Can Be Modulated and Designed on the Basis of Predictions of Nucleosome Occupancy. *ACS Synth. Biol.* **6**, 2086–2095 (2017).
24. Ding, W. *et al.* Engineering the 5' UTR-Mediated Regulation of Protein Abundance in Yeast Using Nucleotide Sequence Activity Relationships. *ACS Synth. Biol.* **7**, 2709–2714 (2018).
25. Shalem, O. *et al.* Systematic dissection of the sequence determinants of gene 3'end mediated expression control. *PLoS Genet.* **11**, e1005147 (2015).
26. Li, J., Liang, Q., Song, W. & Marchisio, M. A. Nucleotides upstream of the Kozak sequence strongly influence gene expression in the yeast *S. cerevisiae*. *J. Biol. Eng.* **11**, 25 (2017).
27. Watson, J. D. *et al.* *Molecular Biology of the Gene*. 6th. ed. (Pearson/Benjamin Cummings, 2008).
28. Koonin, E. V. & Wolf, Y. I. Genomics of bacteria and archaea: the emerging dynamic view of the prokaryotic world. *Nucleic Acids Res.* **36**, 6688–6719 (2008).
29. Lynch, M. & Conery, J. S. The origins of genome complexity. *Science* **302**, 1401–1404 (2003).
30. Pelechano, V., García-Martínez, J. & Pérez-Ortín, J. E. A genomic study of the inter-ORF distances in *Saccharomyces cerevisiae*. *Yeast* vol. 23 689–699 (2006).
31. Zicola, J., Liu, L., Tänzler, P. & Turck, F. Targeted DNA methylation represses two enhancers of FLOWERING LOCUS T in *Arabidopsis thaliana*. *Nat Plants* **5**, 300–307 (2019).
32. Clément, Y., Torbey, P. & Gilardi-Hebenstreit, P. Genome-wide enhancer-gene regulatory maps in two vertebrate genomes. *bioRxiv* (2018).
33. Chepelev, I., Wei, G., Wangsa, D., Tang, Q. & Zhao, K. Characterization of genome-wide enhancer-promoter interactions reveals co-expression of interacting genes and modes of higher order chromatin organization. *Cell Research* vol. 22 490–503 (2012).
34. Mora, A., Sandve, G. K., Gabrielsen, O. S. & Eskeland, R. In the loop: promoter-enhancer interactions and bioinformatics. *Brief. Bioinform.* **17**, 980–995 (2016).
35. Bengio, Y., Courville, A. & Vincent, P. Representation learning: a review and new perspectives. *IEEE Trans. Pattern Anal. Mach. Intell.* **35**, 1798–1828 (2013).
36. Shrikumar, A., Greenside, P. & Kundaje, A. Learning Important Features Through Propagating Activation Differences. *arXiv [cs.CV]* (2017).
37. Shrikumar, A. *et al.* Technical Note on Transcription Factor Motif Discovery from Importance Scores (TF-MoDISco) version 0.5.6.5. *arXiv [cs.LG]* (2018).
38. Ancona, M., Ceolini, E., Öztireli, C. & Gross, M. Towards better understanding of gradient-based attribution methods for Deep Neural Networks. *arXiv [cs.LG]* (2017).
39. Killoran, N., Lee, L. J., DeLong, A., Duvenaud, D. & Frey, B. J. Generating and designing DNA with deep generative models. *arXiv [cs.LG]* (2017).

40. Zeiler, M. D. & Fergus, R. Visualizing and Understanding Convolutional Networks. in *Computer Vision – ECCV 2014* 818–833 (Springer International Publishing, 2014).
41. Alipanahi, B., Delong, A., Weirauch, M. T. & Frey, B. J. Predicting the sequence specificities of DNA- and RNA-binding proteins by deep learning. *Nat. Biotechnol.* **33**, 831–838 (2015).
42. Levo, M. *et al.* Unraveling determinants of transcription factor binding outside the core binding site. *Genome Res.* **25**, 1018–1029 (2015).
43. Rohs, R. *et al.* The role of DNA shape in protein–DNA recognition. *Nature* **461**, 1248–1253 (2009).
44. Slattery, M. *et al.* Absence of a simple code: how transcription factors read the genome. *Trends Biochem. Sci.* **39**, 381–399 (2014).
45. Chiu, T.-P. *et al.* DNASHapeR: an R/Bioconductor package for DNA shape prediction and feature encoding. *Bioinformatics* vol. 32 1211–1213 (2016).
46. Naidoo, T., Sjödin, P., Schlebusch, C. & Jakobsson, M. Patterns of variation in cis-regulatory regions: examining evidence of purifying selection. *BMC Genomics* **19**, 95 (2018).
47. He, B. Z., Holloway, A. K., Maerkl, S. J. & Kreitman, M. Does positive selection drive transcription factor binding site turnover? A test with *Drosophila* cis-regulatory modules. *PLoS Genet.* **7**, e1002053 (2011).
48. Arbiza, L. *et al.* Genome-wide inference of natural selection on human transcription factor binding sites. *Nat. Genet.* **45**, 723–729 (2013).
49. Ludwig, M. Z., Bergman, C., Patel, N. H. & Kreitman, M. Evidence for stabilizing selection in a eukaryotic enhancer element. *Nature* **403**, 564–567 (2000).
50. Wittkopp, P. J. & Kalay, G. Cis-regulatory elements: molecular mechanisms and evolutionary processes underlying divergence. *Nat. Rev. Genet.* **13**, 59–69 (2011).
51. Hahn, M. W. Detecting natural selection on cis-regulatory DNA. *Genetica* **129**, 7–18 (2007).
52. Castillo-Davis, C. I., Hartl, D. L. & Achaz, G. cis-Regulatory and protein evolution in orthologous and duplicate genes. *Genome Res.* **14**, 1530–1536 (2004).
53. Wittkopp, P. J., Haerum, B. K. & Clark, A. G. Evolutionary changes in cis and trans gene regulation. *Nature* **430**, 85–88 (2004).
54. Fay, J. C. & Benavides, J. A. Hypervariable noncoding sequences in *Saccharomyces cerevisiae*. *Genetics* **170**, 1575–1587 (2005).
55. Park, C., Qian, W. & Zhang, J. Genomic evidence for elevated mutation rates in highly expressed genes. *EMBO Rep.* **13**, 1123–1129 (2012).
56. Chen, K., van Nimwegen, E., Rajewsky, N. & Siegal, M. L. Correlating gene expression variation with cis-regulatory polymorphism in *Saccharomyces cerevisiae*. *Genome Biol. Evol.* **2**, 697–707 (2010).
57. Tirosh, I., Reikhav, S., Levy, A. A. & Barkai, N. A yeast hybrid provides insight into the evolution of gene expression regulation. *Science* **324**, 659–662 (2009).
58. Pagán, I., Holmes, E. C. & Simon-Loriere, E. Level of gene expression is a major determinant of protein evolution in the viral order Mononegavirales. *J. Virol.* **86**, 5253–5263 (2012).
59. Subramanian, S. & Kumar, S. Gene expression intensity shapes evolutionary rates of the

- proteins encoded by the vertebrate genome. *Genetics* **168**, 373–381 (2004).
60. de Visser, J. A. G. M. & Krug, J. Empirical fitness landscapes and the predictability of evolution. *Nat. Rev. Genet.* **15**, 480–490 (2014).
 61. Heaton, J. An empirical analysis of feature engineering for predictive modeling. in *SoutheastCon 2016* 1–6 (2016).
 62. Lee, D., Karchin, R. & Beer, M. A. Discriminative prediction of mammalian enhancers from DNA sequence. *Genome Res.* **21**, 2167–2180 (2011).
 63. Li, Y. Establishing glucose- and ABA-regulated transcription networks in Arabidopsis by microarray analysis and promoter classification using a Relevance Vector Machine. *Genome Research* vol. 16 414–427 (2006).
 64. Mejía-Guerra, M. K. & Buckler, E. S. A k-mer grammar analysis to uncover maize regulatory architecture. *BMC Plant Biol.* **19**, 103 (2019).
 65. Anwar, F. *et al.* Pol II promoter prediction using characteristic 4-mer motifs: a machine learning approach. *BMC Bioinformatics* **9**, 414 (2008).
 66. Webb, S. Deep learning for biology. *Nature* **554**, 555–557 (2018).
 67. Kelley, D. R., Snoek, J. & Rinn, J. L. Basset: learning the regulatory code of the accessible genome with deep convolutional neural networks. *Genome Res.* **26**, 990–999 (2016).
 68. Quang, D. & Xie, X. DanQ: a hybrid convolutional and recurrent deep neural network for quantifying the function of DNA sequences. *Nucleic Acids Res.* **44**, e107 (2016).
 69. Zelezniak, A. *et al.* Machine Learning Predicts the Yeast Metabolome from the Quantitative Proteome of Kinase Knockouts. *Cell Syst* **7**, 269–283.e6 (2018).

REVIEWERS' COMMENTS

Reviewer #1 (Remarks to the Author):

The authors have addressed many of the concerns of the past review. I still respectfully disagree with the authors around metrics to determine a successful model. The argument of "everyone else does it" for r^2 etc is not an adequate excuse--you want to lead the field and publish in a top journal? Then lead it and not follow!

On a second point, while I appreciate the use of the leave-some-out approach of the model testing, this is not what I was suggesting. I would want to see how this model would perform on a purely de novo designed element.

These objections notwithstanding, I think the re-delivery of the text and the topic is now suitable for publication and the authors should think about making a discussion about alternative statistics in the text.

Reviewer #2 (Remarks to the Author):

Based on my previous comments, the authors have greatly increased the overall readability of the manuscript by splitting long sentences in shorter ones and providing clearer descriptions of the methodologies used. They have also performed a number of additional analyses and/or have corrected some small oversights. These analyses have further strengthened their conclusions based on their results. Overall this manuscript provides a very interesting analysis and computational model to predict (and explain) gene expression based on the regulatory sequence code.

I have no further comments or remarks and think this is a very nice manuscript.

Responses to reviewer's comments

REVIEWERS' COMMENTS

Reviewer #1 (Remarks to the Author):

Comment 1.1

The authors have addressed many of the concerns of the past review. I still respectfully disagree with the authors around metrics to determine a successful model. The argument of "everyone else does it" for r^2 etc is not an adequate excuse--you want to lead the field and publish in a top journal? Then lead it and not follow!

On a second point, while I appreciate the use of the leave-some-out approach of the model testing, this is not what I was suggesting. I would want to see how this model would perform on a purely *de novo* designed element.

Response 1.1

We thank the Reviewer for the insightful and constructive comments. As we replied previously we *do not* select our models based on R^2 . We have now tested the model on purely *de novo* designed elements. To clarify, *de novo* designed genetic elements are typically random sequences of DNA that are inserted into natural genomic construct-scaffolds that are typically linked to a fluorescent protein as a readout. Using data provided in the study by deBoer et al. 2019¹, comprising measured fluorescence intensities of 9982 randomized promoter constructs, the results show significant correlation (Pearson's $r = 0.507$, p -value $< 1e-16$) between our model predictions and the measured fluorescence levels (Figure R1-1). Despite that we are predicting a completely different readout, i.e. fluorescence levels as opposed to mRNA levels, our model is in good agreement with the experimental data. We thank the Reviewer for encouraging us to perform this analysis and we have included these results as a supplementary figure (Figure S1-5c) and added text in the Discussion section to support our statements that the models demonstrate strong agreement between predicted values and experimental measurements (page 18, lines 608-609).

Figure R1-1. Experimental fluorescence measurements¹ versus predicted expression levels on *de novo* sequence data comprising 9982 randomized promoter constructs within the ANP1 gene scaffold¹. Model trained on *S. cerevisiae* data was used. Red line denotes least squares fit.

Comment 1.2

These objections notwithstanding, I think the re-delivery of the text and the topic is now suitable for publication and the authors should think about making a discussion about alternative statistics in the text.

Response 1.2

We thank the Reviewer for the positive comments. We have added a paragraph to the Discussion about the measures used and alternative statistics in the text (pages 16-17, lines 527-546).

Reviewer #2 (Remarks to the Author):

Comment 2.1

Based on my previous comments, the authors have greatly increased the overall readability of the manuscript by splitting long sentences in shorter ones and providing clearer descriptions of the methodologies used. They have also performed a number of additional analyses and/or have corrected some small oversights. These analyses have further strengthened their conclusions based on their results. Overall this manuscript provides a very interesting analysis and computational model to predict (and explain) gene expression based on the regulatory sequence code.

I have no further comments or remarks and think this is a very nice manuscript.

Response 2.1

We thank the Reviewer for his positive comments and enthusiasm for our work.

References

1. de Boer, C. G. *et al.* Deciphering eukaryotic gene-regulatory logic with 100 million random promoters. *Nat. Biotechnol.* **38**, 56–65 (2020).